# Multi-domain Distribution Learning for De Novo Drug Design

**Arne Schneuing**[1]*, **Ilia Igashov**[1]*, **Adrian W. Dobbelstein**[1], **Thomas Castiglione**[2],
**Michael Bronstein**[3,4] **& Bruno Correia**[1]
[1]École Polytechnique Fédérale de Lausanne, [2]VantAI, Inc., [3]University of Oxford, [4]Aithyra
{arne.schneuing,ilia.igashov,bruno.correia}@epfl.ch

## Abstract

We introduce DRUGFLOW, a generative model for structure-based drug design that integrates continuous flow matching with discrete Markov bridges, demonstrating state-of-the-art performance in learning chemical, geometric, and physical aspects of three-dimensional protein-ligand data. We endow DRUGFLOW with an uncertainty estimate that is able to detect out-of-distribution samples. To further enhance the sampling process towards distribution regions with desirable metric values, we propose a joint preference alignment scheme applicable to both flow matching and Markov bridge frameworks. Furthermore, we extend our model to also explore the conformational landscape of the protein by jointly sampling side chain angles and molecules.

## 1 Introduction

Small molecules are the predominant class of FDA-approved drugs with a share of 85%, and more than 95% of known drugs target human or pathogen proteins (Santos et al., 2017). At the same time, the cost and duration of the development of new drugs are skyrocketing (Simoens & Huys, 2021). This sparks increasing interest in the computational design of small molecular compounds that bind specifically to disease-associated proteins and thus reduce the amount of costly experimental testing.

In recent years, the machine learning community has contributed a plethora of generative tools addressing drug design from various angles (Du et al., 2024). Some methods directly optimize specific drug properties, using techniques such as reinforcement learning (Popova et al., 2018; Gottipati et al., 2020) or search-based approaches (Gómez-Bombarelli et al., 2018; Swanson et al., 2024). However, these methods typically require careful tuning of the objective function to avoid exploiting imperfect computational oracles and overly maximizing one desired property (e.g. binding affinity) at the expense of another (e.g. oral bioavailability). Additionally, one often aims to design a suitable 3D binding pose along with the chemical structure of the molecule, which substantially increases the degrees of freedom. Many optimization algorithms struggle to efficiently navigate such vast design spaces.

Following a different approach, probabilistic generative models learn to generate drug-like molecules directly from data (Hoogeboom et al., 2022; Vignac et al., 2022). Here, the design objectives are implicitly encoded in the training data set. While these methods may not outperform direct optimization on isolated metrics, they are well suited for the multifaceted nature of drug design as they learn "what a drug looks like" in a more general way. Once trained on sufficient high-quality data, these models can capture a more holistic picture of the molecular space compared to models optimized for a limited set of target metrics.

The strength of generative modeling lies in its ability to reproduce patterns seen in the training data. However, many prior works on generative molecular design have focused on absolute metric values in baseline comparisons (e.g. identifying the best docking scores or synthetic accessibility estimates), even though the proposed models did not directly optimize these quantities. For example, Table 6 from the recent survey on generative structure-based drug design methods by Zhang et al. (2023b) highlights a common evaluation strategy used in this context. We argue that the evaluation

---

*These authors contributed equally

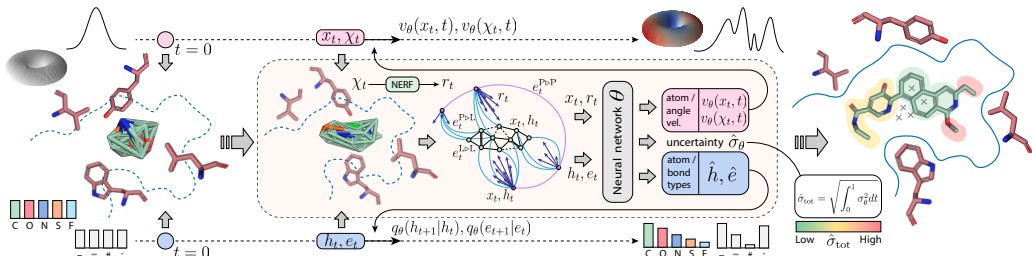

Figure 1: **Method overview.** DRUGFLOW operates on continuous ligand atom coordinates $x_t$, discrete atom types $h_t$, and bond types $e_t$. Its extension FLEXFLOW additionally operates on continuous side chain angles $\chi_t$. As shown on the left, coordinates and angles are sampled from a Gaussian prior in 3D and a uniform prior on the torus, respectively. Discrete types are sampled from categorical prior distributions (uniform or marginal). Denoising schemes for continuous and discrete data types are based on conditional flow matching (top) and Markov bridge models (bottom). As shown in the middle, at time step $t$, we process the noisy data to obtain the 3D graph: side chain angles are transformed into vector features pointing to atom positions using NERF, and three types of edges are introduced using different distance cutoffs—edges between ligand atoms $e_t^{L \triangleright L}$, edges between ligand and protein $C_\alpha$ atoms $e_t^{P \triangleright L}$, and edges between protein $C_\alpha$ atoms $e_t^{P \triangleright P}$. The graph is processed by a neural network $\theta$ which outputs velocities $v_\theta(x_t, t)$, $v_\theta(\chi_t, t)$ and type predictions $\hat{h}_t$, $\hat{e}_t$ which are fed into the generative modeling framework. The model can adapt the size of the molecule during sampling and label excessive atoms that will be eventually removed (shown as crosses on the right). Additionally, DRUGFLOW outputs per-atom uncertainty values $\hat{\sigma}_{\text{tot}}$.

of generative models for drug design should instead be centered around their ability to represent the training distribution accurately – analogous to how image generation methods are typically assessed (Heusel et al., 2017). Without additional fine-tuning or sampling strategies, it is unreasonable to expect a model to substantially improve any score compared to the training set.

In this work, we present DRUGFLOW, a new generative model for structure-based drug design, that simultaneously learns the distribution of protein-binding molecules in three data domains. DRUGFLOW generates discrete atom and bond types as well as atom coordinates in the Euclidean space. Its extended version, FLEXFLOW, additionally samples side chain configurations of the binding pocket represented as angles on a hypertorus. This allows us to sample probabilistic ensembles of possible binding modes and enables drug design for targets in unbound conformations. Both DRUGFLOW and FLEXFLOW are conditioned on fixed protein backbone coordinates and amino acid types, which are used as context for denoising. We further introduce a virtual node type to allow the model to dynamically add or remove atoms and thus learn about the distribution of ligand sizes rather than requiring it to be pre-specified. Finally, we add an uncertainty head to our model that is trained in an end-to-end fashion to identify out-of-distribution samples and rank molecules at inference time.

Based on our observations, we focus our evaluation primarily on the distribution learning capabilities of the proposed generative model, comparing it to established baselines. To this end, we assess molecular properties and structural features using distance functions between distributions derived from generated samples and training data points, and demonstrate that DRUGFLOW molecules closely match the data distribution across a broad range of metrics. This suggests that DRUGFLOW can be retrained on curated datasets to steer the generation of samples towards desired regions of the chemical space for various practical applications. However, recognizing that excessive filtering may lead to insufficient training data, we present an alignment strategy which allows us to update a pre-trained model based on user preferences.

We summarize the main contributions of this work as follows.

**Conceptual novelty**  We present DRUGFLOW and FLEXFLOW, new generative models for structure-based drug design, and introduce three conceptually new features: (1) and end-to-end trained uncertainty estimate that successfully detects out-of-distribution samples, (2) an adaptive size selection method that discards excessive atoms during sampling, and (3) a protein conformation sampling module that samples realistic side chain rotamers.

**Performance** We propose to evaluate generative models in a way that better captures their training objective, and use the new benchmarking framework to show that DRUGFLOW is a state-of-the-art distribution learner for structure-based drug design.

**Practical relevance** Recognizing that medicinal chemists ultimately aim to optimize molecules for specific design objectives, we implement a preference alignment scheme that allows us to efficiently sample molecules with improved target properties. Recognizing that medicinal chemists ultimately aim to optimize molecules for specific design objectives, we implement a preference alignment scheme that allows us to efficiently sample molecules with improved target properties.

## 2 METHODS

Our base model is a probabilistic model operating simultaneously on atom types, bond types and coordinates, thereby combining generative processes for discrete and continuous data types. We use Euclidean flow matching (Lipman et al., 2022) for ligand coordinates and combine it with Markov bridge models (Igashov et al., 2023) applied to atom and bond types to generate the discrete molecular graphs. More background on these generative modeling frameworks is presented in Appendix A.1.

The backbone model is a heterogeneous graph neural network that has independent trainable weights for ligand and protein node types, as well as ligand, protein, and interaction edge types. Each protein node represents a whole residue but we include vector-valued input features to encode the locations of all atoms belonging to the residue. This allows us to reduce computational complexity while preserving full atomic detail. The neural network predicts several node-level and edge-level outputs required for sampling, including logits for atom and bond types and vectors for the coordinates. All trainable operations applied to geometric quantities are implemented as geometric vector perceptrons (GVPs) (Jing et al., 2020) to ensure that predicted vector fields transform equivariantly. Full architectural details are provided in Appendix A.5.

Below, we describe our conceptual novelties one by one, starting with uncertainty estimation in Section 2.1, followed by ligand size adaptation in Section 2.2 and protein flexibility in Section 2.3. Finally, we present a fine-tuning approach to align the general distribution learner with user specified preferences in Section 2.4. Our method is schematically depicted in Figure 1.

### 2.1 UNCERTAINTY ESTIMATION

To identify out-of-distribution (OOD) samples and endow DRUGFLOW with an intrinsic uncertainty estimate, we rely on a technique that has been successfully used for regression problems in the past (Nix & Weigend, 1994; Lakshminarayanan et al., 2017). We assume that the flow matching regression error is normally distributed with standard deviation $\sigma_\theta$, and derive a loss function that maximises the likelihood of the true vector field under this uncertainty model,

$$\mathcal{L}_{\text{FM-OOD}} = \mathbb{E}_{t,q(\boldsymbol{x}_1),p(\boldsymbol{x}_0)} \; \frac{d}{2} \log \sigma_\theta^2(\boldsymbol{x}_t, t) + \frac{1}{2\sigma_\theta^2(\boldsymbol{x}_t, t)} \|\boldsymbol{v}_\theta(\boldsymbol{x}_t, t) - \dot{\boldsymbol{x}}_t\|^2 + \frac{\lambda}{2} |\sigma_\theta^2(\boldsymbol{x}_t, t) - 1|^2, \quad (1)$$

where $\boldsymbol{v}_\theta(\boldsymbol{x}_t, t) \in \mathbb{R}^d$ and $\sigma_\theta(\boldsymbol{x}_t, t) \in \mathbb{R}$ are two output heads of the neural network and $\dot{\boldsymbol{x}}_t$ is the ground-truth conditional vector field. The derivation is provided in Appendix A.2.

A model trained in this way provides us with a per-atom uncertainty score in addition to the vector field for flow matching at every sampling step. We integrate the step-wise score to obtain an uncertainty value for the entire sampling trajectory (motivated in Appendix A.3) as

$$\hat{\sigma}_{\text{tot}} = \sqrt{\int_0^1 \sigma_\theta^2(\boldsymbol{x}_t, t) dt}, \quad (2)$$

and assign this value to the resulting generated atom.

### 2.2 END-TO-END SIZE ESTIMATION

It is common practice to choose molecule sizes for diffusion and flow matching models in structure-based drug design *a priori*. The number of generated atoms is often either a user-specified hyper-parameter, sampled from the empirical distribution of sizes in the training set (Hoogeboom et al.,

2022; Schneuing et al., 2022; Guan et al., 2023a), or estimated by a separate neural network (Igashov et al., 2022). This approach, however, prevents the generative neural network from adapting to the context as the current sample evolves. For instance, if initially too many atoms are specified it might be impossible for the model to create a molecule without steric clashes with the surrounding protein atoms.

In order to also learn this aspect of the data distribution in an end-to-end manner, we aim to adapt the molecule size during the generative process, which effectively changes the dimension of the modeled system (Campbell et al., 2024). To do so, we introduce a virtual ("no atom") node type the model can sample. All nodes of this type are treated as completely disconnected (i.e. all bonds of virtual nodes have "None" type) and will be removed at the end of sampling. During training, we add $n_{\text{virt}} \sim U(0, N_{\text{max}})$ virtual nodes to each training sample. In order to treat virtual nodes and atoms in the same way, we need to attach coordinates to them as well. While different design choices are possible, we find that placing them in the center of mass of the ligand works well in practice as it provides a clear reference point for the network to regress toward. Note that this approach still requires pre-specifying the number of *nodes in the computational graph* which serves as an upper bound for the number of atoms in the generated molecule.

## 2.3 PROTEIN FLEXIBILITY

Incorporating the dynamics and flexibility of protein structures is one of the key open challenges for structure-based drug design (Fraser & Murcko, 2024). As a first step to addressing scenarios in which the bound structure of the target protein is unknown or assuming a single static pocket configuration is too restrictive, we extend our base model to also generate side chain torsion angles for all pocket residues while sampling new ligands, thereby permitting full side chain flexibility.

To this end, we apply flow matching on the hypertorus that describes all torsion angles. Following the Riemannian Flow Matching framework of Chen & Lipman (2023), we approximate vector fields in the tangent space. More details and definitions for flow matching on Riemannian manifolds is provided in Appendix A.1. Similar to related works on flow matching for angular domains (Yim et al., 2023a; Lee & Kim, 2024) we find a non-linear scheduler $\kappa(t)$, which controls how quickly the geodesic distance between start and end point of a trajectories decreases, to be beneficial to sample quality. However, unlike these works we adopt a polynomial scheduler $\kappa(t) = (1 - t)^k$ with $k = 3$. This has a similar effect on the sampling trajectories as the exponential scheduler $\kappa(t) = e^{-ct}$ if $c = 5$ but strictly fulfills the theoretical requirement on the boundary conditions: $\kappa(0) = 1$ and $\kappa(1) = 0$. Using the exponential and logarithm maps associated with the manifold, we derive the following updated flow and vector fields:

$$\boldsymbol{x}_t = \exp_{\boldsymbol{x}_0} \left( (1 - (1 - t)^k) \log_{\boldsymbol{x}_0}(\boldsymbol{x}_1) \right), \tag{3}$$

$$\dot{\boldsymbol{x}}_t = k(1 - t)^{k-1} \log_{\boldsymbol{x}_0}(\boldsymbol{x}_1). \tag{4}$$

We use this scheduler both for training and sampling.

While the generative process is performed entirely in angular space to enforce physical plausibility, we present the full-atomic information more explicitly to the neural network and convert the side chain dihedral angles back to atom positions in every training and sampling step. This operation is performed efficiently using the Natural Extension Reference Frame (NERF) algorithm (Parsons et al., 2005; Alcaide et al., 2022) in parallel for each residue.

## 2.4 MULTI-DOMAIN PREFERENCE ALIGNMENT

Many real-world applications require generating molecules with properties underrepresented in the initial training data. To address this need, we employ an alignment scheme inspired by Direct Preference Optimization (DPO), a technique originally used to align large language models (Rafailov et al., 2023) and later adapted to diffusion models (Wallace et al., 2024).

Using a pre-trained *reference* model $\theta$, we generate a synthetic dataset of preference pairs $\mathcal{D} = \{(x_i^w, x_i^l)\}_i$, where the *winning* samples $x_i^w$ have more desirable molecular properties compared to the *losing* samples $x_i^l$ (see Section 3.5 for details on our preference datasets). To align our method with these preferences, we introduce another (*aligned*) model $\varphi$, initialised with the weights of the

Table 1: Wasserstein distance between marginal distributions of continuous molecular data (bond distances and angles), drug-likeness (QED), synthetic accessibility (SA), lipophilicity (logP) and numbers of rotatable bonds (RB). The last column reports the Jensen-Shannon divergence between the joint distributions of four molecular properties (QED, SA, logP and Vina efficiency score). The best result is highlighted in bold, the second best is underlined.

| | Top-3 bond distances | | | Top-3 bond angles | | | Molecular properties | | | | |
|---|---|---|---|---|---|---|---|---|---|---|---|
| Method | C–C | C–N | C=C | C–C=C | C–C–C | C–C–O | QED | SA | logP | RB | $\text{JSD}_{\text{all}}$ |
| POCKET2MOL | 0.050 | 0.024 | 0.045 | 2.173 | 2.936 | 3.938 | 0.072 | 0.576 | 1.209 | 2.861 | 0.223 |
| DIFFSBDD | 0.041 | 0.039 | 0.042 | 3.632 | 8.166 | 7.756 | 0.065 | 1.570 | 0.774 | 0.928 | 0.274 |
| TARGETDIFF | 0.017 | 0.019 | 0.028 | 4.281 | 3.422 | 4.125 | 0.050 | 1.518 | **0.489** | 0.354 | 0.242 |
| DRUGFLOW | **0.017** | **0.016** | **0.016** | **0.952** | **2.269** | **1.941** | **0.014** | **0.317** | 0.665 | **0.144** | **0.099** |

Table 2: Wasserstein distance between distributions of binding efficiency scores and normalized numbers of different protein-ligand interactions. The best result is highlighted in bold, the second best is underlined.

| | Binding efficiency | | Protein-ligand interactions | | | | | |
|---|---|---|---|---|---|---|---|---|
| Method | Vina | Gnina | H-bond | H-bond (acc.) | H-bond (don.) | $\pi$-stacking | Hydrophobic |
| POCKET2MOL | 0.064 | 0.044 | 0.040 | 0.026 | 0.014 | 0.007 | **0.027** |
| DIFFSBDD | 0.086 | 0.043 | 0.047 | 0.030 | 0.017 | 0.011 | 0.044 |
| TARGETDIFF | 0.034 | 0.030 | 0.031 | 0.021 | 0.010 | 0.012 | 0.039 |
| DRUGFLOW | **0.028** | **0.013** | **0.019** | **0.012** | **0.007** | **0.006** | 0.036 |

reference model $\theta$. From now on, the reference model $\theta$ is no longer optimised, and further training is only performed on the new model $\varphi$.

For each data domain $c \in \{\text{coord}, \text{atom}, \text{bond}\}$, we compute the loss terms $\mathcal{L}_c^w(\varphi) := \mathcal{L}_c(x^w, \varphi)$ and $\mathcal{L}_c^l(\varphi) := \mathcal{L}_c(x^l, \varphi)$ for winning and losing samples, respectively. More specifically, these are the flow matching loss for coordinates (Eq. 12) and the Markov bridge model loss for atom and bond types (Eq. 24). Additionally, we calculate the corresponding loss terms $\mathcal{L}_c^w(\theta)$ and $\mathcal{L}_c^l(\theta)$ using the reference model $\theta$ with fixed parameters. Using individually weighted loss differences $\Delta_c^w = \mathcal{L}_c^w(\varphi) - \mathcal{L}_c^w(\theta)$ and $\Delta_c^l = \mathcal{L}_c^l(\varphi) - \mathcal{L}_c^l(\theta)$, we define the multi-domain preference alignment (MDPA) loss as follows,

$$\mathcal{L}_{\text{MDPA}}(\varphi) = -\log \sigma\left(-\beta_t \sum_c \lambda_c \left(\Delta_c^w - \Delta_c^l\right)\right) + \lambda_w \mathcal{L}^w(\varphi) + \lambda_l \mathcal{L}^l(\varphi), \tag{5}$$

where $\lambda_c, \lambda_w, \lambda_l$ are adjustable weights, and $\sigma$ is the sigmoid function. Note that we regularize training by adding the overall loss terms $\mathcal{L}^w(\varphi)$ and $\mathcal{L}^l(\varphi)$ (Eq. 25) for the winning and losing samples, respectively. We find that such a regularization significantly enhances training stability. More details are provided in Appendix A.4.

## 3 EXPERIMENTS

### 3.1 MULTI-DOMAIN DISTRIBUTION LEARNING

In this work, we focus on the generative capabilities of our model, namely on its ability to learn the training data distribution. While a common trend in the community is to report absolute values of various molecular properties and docking scores, we stress that such an evaluation is relevant only for methods whose primary goal is property optimisation, such as preference alignment (Cheng et al., 2024) or optimisation in the latent space (Gómez-Bombarelli et al., 2018). Unless stated specifically, DRUGFLOW **does not optimise for any specific property and aims to learn the data distribution only**. Therefore, instead of absolute values of molecular properties we measure the proximity of distributions of these properties computed on the generated samples and the training data. As we show in Section 3.5, absolute values of the target metrics can be increased by fine-tuning on relevant data or using the preference alignment scheme.

**Metrics** We compute Jensen-Shannon divergences for the categorical distributions of atom types, bond types and ring systems (Walters, 2022; 2021). We use the Wasserstein-1 distance for the bond length distributions of the three most common bond types (C–C, C–N and C=C), the three most common bond angles (C–C=C, C–C–C and C–C–O) as well as the number of rotatable bonds per molecule. We also apply the Wasserstein distance to computational scores relevant to applications in medicinal chemistry: Quantitative Estimate of Drug-likeness (QED) (Bickerton et al., 2012), Synthetic Accessibility (SA) (Ertl & Schuffenhauer, 2009) and lipophilicity (logP) (Wildman & Crippen, 1999). Binding efficiency, defined as a computational binding affinity score divided by the number of atoms in the molecule, is assessed in the same manner. We report efficiency scores instead of affinity scores due to the high correlation of the latter with molecule size (Cremer et al., 2024). We use both Vina and Gnina docking scores (McNutt et al., 2021) as binding affinity oracles. Besides, we compare normalized counts of various types of non-covalent interactions as detected by ProLIF (Bouysset & Fiorucci, 2021). To do this, we divide the number of interactions by the number of atoms in each molecule and compute Wasserstein distances to compare the resulting distributions. Finally, we also employ the Fréchet ChemNet Distance (FCD) (Preuer et al., 2018) to assess how well the model approximates the training data distribution.

**Dataset & Baselines** We use the CrossDocked dataset (Francoeur et al., 2020) with $100\,000$ protein-ligand pairs for training and 100 proteins for testing, following previous works (Luo et al., 2021; Peng et al., 2022). The data split was done by 30% sequence identity using MMseqs2 (Steinegger & Söding, 2017). Ligands that do not pass all PoseBusters Buttenschoen et al. (2024) filters were removed from the training set. We compare DRUGFLOW with an autoregressive method, POCKET2MOL (Peng et al., 2022), and two diffusion-based methods, TARGETDIFF (Guan et al., 2023a) and DIFFSBDD (Schneuing et al., 2022). We generated 100 samples for each test set target with DRUGFLOW and selected only molecules that passed the RDKit validity filter.

**Results** The distances between sampling distributions measured on the various discrete and continuous characteristics of molecules and interactions are summarized in Tables 1, 2 and 3. DRUGFLOW shows convincing all-round performance and outperforms other methods in almost all aspects. POCKET2MOL achieves slightly better results on bond types and hydrophobic interactions. TARGETDIFF gets the best results on the lipophilicity metric. However, in these few cases DRUGFLOW consistently ranks a close second. Along with the marginal distributions of various geometric and

Table 3: Fréchet ChemNet Distance and Jensen-Shannon divergence between distributions of discrete molecular data. The best result is highlighted in bold, the second best is underlined.

| Method | FCD | Atoms | Bonds | Rings |
|---|---|---|---|---|
| POCKET2MOL | 12.703 | 0.081 | **0.044** | 0.446 |
| DIFFSBDD | 11.637 | 0.050 | 0.227 | 0.588 |
| TARGETDIFF | 13.766 | 0.076 | 0.240 | 0.632 |
| DRUGFLOW | **4.278** | **0.043** | 0.060 | **0.391** |

chemo-physical characteristics, we compare joint distributions of the molecular properties. To do this, we created histograms of the joint distributions of QED, SA, logP and Vina efficiency scores (10 bins per score, Figure 21) and compute the Jensen-Shannon divergence between them. DRUGFLOW outperforms other methods by a large margin, as demonstrated by $\text{JSD}_{\text{all}}$ in Table 1. Additionally, our method substantially outperforms other baselines in FCD. To further analyse this result, we applied Principal Component Analysis (PCA) to reduce the dimensionality of the ChemNet embeddings and visualized the first two principal components on a 2D plane. As shown in Appendix Figure 7, DRUGFLOW covers considerably more modes of the training distribution compared to the other methods. Distributions of other metrics are provided in Appendix B.1. Additionally, we compute novelty, uniqueness, and overall quality of the samples using PoseBusters (Buttenschoen et al., 2024). As shown in Appendix Table 6, DRUGFLOW demonstrates competitive results in all these metrics as well. Appendix Tables 7, 8, and 9 provide additional results for different variations of DRUGFLOW and FLEXFLOW. The statistical significance is ensured in Appendix B.10.

## 3.2 OUT-OF-DISTRIBUTION DETECTION

The uncertainty estimation technique described in Section 2.1 provides per-atom uncertainty scores, as shown in Figure 2A. Here, we showcase six molecules of the same size with the highest and lowest global (i.e. averaged over atoms) uncertainty scores.

To demonstrate the ability of our uncertainty score to detect out-of-distribution samples, we plotted histograms of several basic geometric properties of the generated molecules. The histogram bins

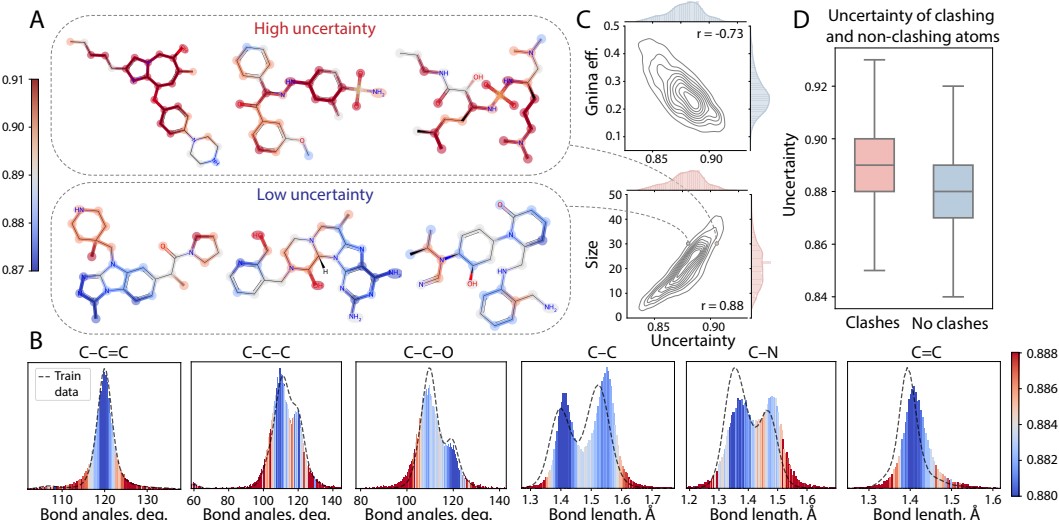

Figure 2: **Uncertainty estimate detects out-of-distribution samples and correlates with various molecule characteristics.** (A) Examples of samples with the same size (30 heavy atoms) and high and low global uncertainties. Each atom is highlighted according to its local uncertainty score. (B) Distributions of top-3 bond angles and bond lengths in DRUGFLOW samples (histograms) and training data (dashed lines). The histogram bins are color-coded according to the average uncertainty values of the corresponding data points. In all cases, samples from the distribution tails have high uncertainty. Main modes of the distributions have lower uncertainty on average. The color bar is scaled for a better visibility. (C) Correlation of the global uncertainty score with Gnina efficiency score and size of the molecules. (D) Atoms clashing with the protein have higher uncertainty scores than non-clashing atoms.

are color-coded according to the average uncertainty values of the corresponding data points. As shown in Figure 2B, the model tends to assign high uncertainty to data points in the tails of the distributions, while sampling from the central regions (modes) with high confidence. The slight limitation is that most of the values fall within the range of 0.85 to 0.92. However this does not affect the discriminative ability of the score.

Furthermore, our analysis reveals strong correlations between global uncertainty and both molecule size and docking efficiency, as shown in Figure 2C. While the correlation with molecule size is expected, the correlation with size-agnostic ligand efficiency scores is particularly interesting and has practical implications. We hypothesize that this correlation may be a byproduct of an efficient distribution learning process, wherein, among other data aspects, the model learns to avoid steric clashes between protein and ligand atoms. Indeed, as shown in Figure 2D, atoms sampled without clashes tend to have lower uncertainty values compared to clashing atoms.

### 3.3 DISTRIBUTION OF MOLECULES SIZES

As explained in Section 2.2, at every training step we add to the molecule a random number of virtual nodes sampled from $U(0, N_{max})$. It means that on average the model was trained to label as virtual (i.e. to remove) $N_{max}/2$ atoms per molecule. In our experiments, we set $N_{max} = 10$, and therefore expect the model to remove on average 5 atoms during sampling. To test this hypothesis, we sampled molecules with ground-truth sizes and 5 atoms added on top. As shown in Figure 3C, the model indeed tends to remove about 5 atoms. The deviation from the uniform training distribution (shown in red) is evidence of the model's ability to learn the conditional distribution of molecule sizes given pockets. CrossDocked pockets are defined based on a distance cutoff, which introduces a dependency between ligand and pocket sizes.

Next, we studied the ability of the model to minimise steric clashes with the pocket when a large number of (computational) nodes is provided. To maximise the number of geometric constraints, we selected a protein with a deeply buried pocket (PDB: 1L3L) and a tightly bound ligand (17 atoms),

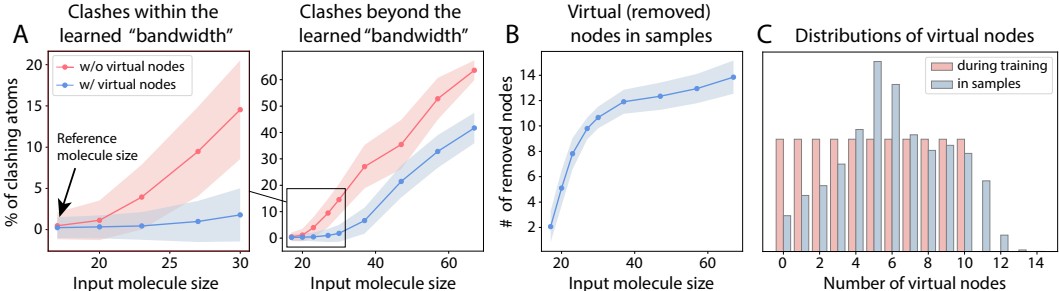

Figure 3: **DRUGFLOW learns the conditional size distribution of molecules given protein pockets.** (A) Our model effectively removes redundant atoms to avoid clashes. (B) Trained with maximum $N_{\max} = 10$ virtual nodes, the model struggles to remove much more atoms. (C) Even though the training distribution of number of added virtual nodes is $U(0, N_{\max})$, the distribution of the removed nodes during sampling suggests that the model learned the dependency between ligand and pocket sizes.

as shown in Appendix Figure 16. For different input sizes (between 17 and 67 atoms), we sampled 1000 molecules per each size value and measured the percentage of clashing atoms and numbers of atoms removed by the model. We repeated the same experiment with a similar model trained without virtual nodes. As shown in Figure 3A, the model produces minimum clashes as long as the number of excessive atoms does not exceed 10, the "bandwidth" the model was trained with. Higher numbers are out of the training distribution, and therefore the model fails to remove more, as shown in Figure 3B. We believe that scaling up the maximum number of virtual nodes during training will enable the model to operate in a fully adaptive size selection regime.

### 3.4 LEARNING THE DISTRIBUTION OF SIDE CHAIN ROTAMERS

Here we evaluate how well FLEXFLOW, the version of our model that simultaneously generates side chain conformations, recovers the distribution of bound side chain rotamers. Because all proteins in our test set are provided in ligand-bound form, we repacked their side chains in absence of the small molecule to approximate their unbound structures. We relaxed side chains with the Rosetta repack protocol (Conway et al., 2014) and achieved an RMSD of about 1.98Å for side chain heavy atoms (Figure 4A, middle).

**Recovering bound configurations** As a first test, we sampled 20 sets of side chain torsion angles per test set target with FLEXFLOW while keeping the ligand fixed. We achieve this by using the ground truth vector field and transition probabilities instead of the predicted quantities for all ligand-related variables. Figure 4A shows that the model samples pocket structures close to the original bound conformations (with median side chain of RMSD 1.75Å). This decrease in RMSD compared to the unbound pocket is expected because fixing the ligand binding pose constrains the space of feasible solutions. For comparison, we also include the distribution of RMSD values that results from simply taking random angles from the prior distribution (Figure 4, left).

**Distribution of side chain angles** Next, we freely generate molecules and bound configurations of the protein-ligand complex and assess whether the resulting side chain rotamers are in accordance with the bound structures from the training set. Starting from a completely random prior, FLEXFLOW manages to recover the rotameric modes of the reference structures accurately. Figure 4B shows the distributions of the first two side chain torsion angles of three bulky amino acids. Analogous plots for all 14 amino acids with at least two side chain angles are presented in Appendix Figure 17.

### 3.5 PROPERTY OPTIMIZATION WITH PREFERENCE ALIGNMENT

We conduct preference alignment with respect to four molecular properties: QED, SA, Vina efficiency score, and Rapid Elimination of Swill (REOS) filters (Walters et al., 1998). REOS filters include various structural alerts designed to detect problematic compounds in screening libraries (Baell & Holloway, 2010; Hann et al., 1999; Pearce et al., 2006). For REOS, molecules are classified as

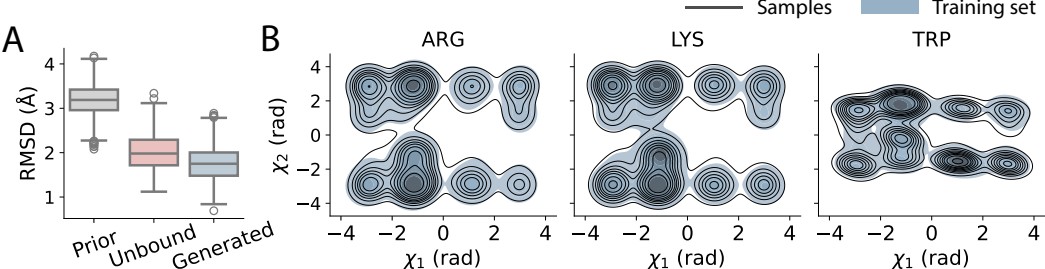

Figure 4: **FLEXFLOW samples realistic side chain conformations.** (A) Side chain root-mean-square deviation (RMSD) for random samples from the model's prior (left), relaxed pockets in absence of the ligand (middle), and FLEXFLOW-generated side chain conformers for a fixed ligand structure (right). (B) Distributions of $\chi_1$ and $\chi_2$ angles for Arginine, Lysine and Tryptophan. We compare FLEXFLOW samples to the bound pocket conformations from the training set.

either passing all REOS filters (winning) or failing at least one (losing). For continuous metrics, we set thresholds requiring the winning sample to outperform the losing one by at least $0.5$ for SA, and $0.1$ for both QED and Vina efficiency. In each pair, both molecules are generated for the same pocket. In addition, we curate a dataset where winning samples surpass losing samples across all four properties. To perform preference alignment, we initialize the model with the reference DRUGFLOW model parameters and train until convergence of the loss $\mathcal{L}_{\mathrm{MDPA}}$ (Eq. 5). We then select a checkpoint using the validity metric on the validation set. To compare our preference alignment method against a simpler optimization strategy, we fine-tune the same reference model on the winning samples only.

Figure 5 shows the performance gains of the aligned and fine-tuned models compared to the training data and the reference model. Additionally, we visualize distribution shifts of the target metrics in Appendix Figure 18 and show their statistical significance in Figure 19. Our results indicate that the preference-aligned models consistently exceed their fine-tuned counterparts across all metrics. However, these improvements are achieved at the cost of moderately reduced molecular validity (Appendix Table 13). Notably, the model optimised for all four properties at once demonstrates competitive results across all target metrics.

## 4 RELATED WORK

**Generative models for molecule generation** This paper builds on a large body of work on probabilistic models for molecule generation. Some of these models generate molecules unconditionally without knowledge of the target protein. They either create molecular graphs (Jo et al., 2022; Vignac et al., 2022), 3D atomic point clouds (Gebauer et al., 2019; Garcia Satorras et al., 2021; Hoogeboom et al., 2022) or both Vignac et al. (2023). More closely related to the work presented here, another family of models attempts to generate novel chemical matter conditioned on three-dimensional context, typically a structural model of a target protein. Among these, most of the models sample atom positions and types without providing explicit information about covalent bonds (Liu et al., 2022; Ragoza et al., 2022; Schneuing et al., 2022; Igashov et al., 2022; Guan et al., 2023a; Lin et al., 2022; Xu et al., 2023). Others generate the full molecular graph structure and binding pose jointly (Peng et al., 2022; Guan et al., 2023b; Zhang et al., 2023a). Notably, most recent 3D models belong to the family of diffusion probabilistic models (Schneuing et al., 2022; Guan et al., 2023a; Lin et al., 2022; Xu et al., 2023; Guan et al., 2023b; Weiss et al., 2023) or follow the related flow matching paradigm (Song et al., 2024; Dunn & Koes, 2024; Irwin et al., 2024). So far, it is uncommon for these models to handle varying dimensionality (i.e. molecule sizes) (Campbell et al., 2024) or incorporate protein flexibility although torsional flow matching has already been applied to protein side chain packing (Lee & Kim, 2024) and peptide design (Lin et al., 2024).

**Confidence prediction** Neural confidence estimates are crucial components of many popular methods for biomolecular applications (Jumper et al., 2021; Corso et al., 2022; Abramson et al., 2024). Typically, such estimates are obtained by training a separate neural network or auxiliary output head to approximate the prediction error of the main model either during training (Jumper et al., 2021;

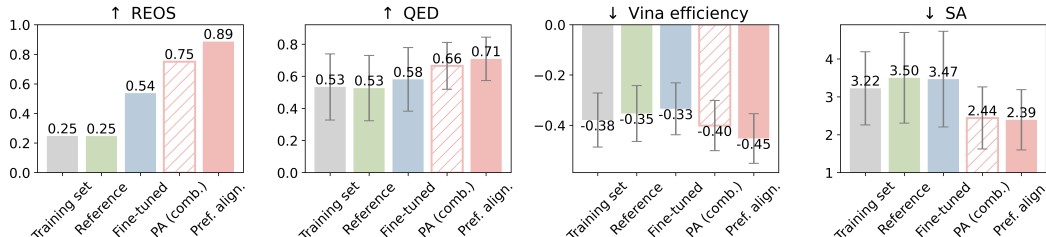

Figure 5: **Preference alignment improves target properties.** Comparison of models across four molecular properties: REOS, QED, SA, and Vina efficiency. Bars represent the training set (gray), reference model (green), fine-tuned model (blue), preference-aligned model with combined preferences (dashed), and preference-aligned model for the metric specified above each plot (red). Preference-aligned models consistently outperform fine-tuning, with the best results in the metric-specific models. Results are based on 100 sampled molecules per target.

Abramson et al., 2024) or for samples from a trained model (Corso et al., 2022). Unlike our approach, this requires minimizing a separate secondary loss function. Furthermore, to enable this form of confidence estimation based on final outputs during training, diffusion (or flow matching) models like AlphaFold 3 (Abramson et al., 2024) must perform expensive rollout schemes to regress their output errors directly. Here, we circumvent this by deriving a global uncertainty score from local, step-wise predictions.

**Preference alignment** Generating molecules with desired properties is a critical step in applying generative models to biomolecules. Existing methods, such as reinforcement learning (RL)-based approaches (Zhou et al., 2019) or GANs (Maziarka et al., 2020), often involve a two-step process: first, to generate molecules, then optimize them. In contrast, preference alignment techniques allow generative models to directly produce outputs aligned with human or domain-specific preferences in a single step. DPO offers a key advantage over RL by eliminating the need to learn a reward function, making it a less exploitable way to optimize preferences (Rafailov et al., 2023). DPO has been applied to fine-tune chemical language models (Park et al., 2023) and its adaptation for diffusion models (Wallace et al., 2024) has already found applications in antibody design (Zhou et al., 2024) and structure-based drug design (Cheng et al., 2024; Gu et al., 2024). In DecompDPO (Cheng et al., 2024), molecules are decomposed into fragments, allowing preference optimization at both local and global levels, which helps resolve conflicting preferences between molecular substructures. In AliDiff (Gu et al., 2024), the DPO objective is decomposed across atom types and coordinates and rescaled according to reward differences in preference pairs. Our approach extends this by applying a preference alignment scheme simultaneously in different generative frameworks (continuous flow matching and discrete Markov bridge models).

## 5 CONCLUSION

In this work, we advocate for a distribution learning-centered evaluation of generative models for drug design. Methods that perform well in this framework, can later improve their task-specific performance through curated datasets, advanced sampling strategies, or fine-tuning to better align with user preferences. We introduce DRUGFLOW and demonstrate that it consistently achieves state-of-the-art distribution learning performance across various orthogonal metrics. DRUGFLOW also learns the distribution of molecular sizes and is able to detect out-of-distribution samples. Besides, its extension, FLEXFLOW, additionally operates on protein residues and learns the distribution of side chain conformations. Finally, we discuss a preference alignment strategy that allows us to sample molecules with improved properties, and show its effectiveness across four different metrics.

## REPRODUCIBILITY STATEMENT

All methodological details are described in Section 2 and Appendix A. Source code is available at `https://github.com/LPDI-EPFL/DrugFlow`.

## ACKNOWLEDGMENTS

The authors would like to thank Rebecca Neeser, Julian Cremer, Charles Harris, Felix Dammann, Yuanqi Du, Evgenia Elizarova, and Simon Crouzet for their helpful feedback. This work was supported by the Swiss National Science Foundation grant 310030_197724. I.I. has received funding from the European Union's Horizon 2020 Research and Innovation Programme under the Marie Sklodowska-Curie grant agreement No. 945363. A.W.D. was supported by the Studienstiftung des deutschen Volkes. M.B. is partially supported by the EPSRC Turing AI World-Leading Research Fellowship No. EP/X040062/1 and EPSRC AI Hub No. EP/Y028872/1.

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

# Appendix for
# "Multi-domain Distribution Learning
# for De Novo Drug Design"

TABLE OF CONTENTS

# A  EXTENDED METHODS

## A.1  GENERATIVE FRAMEWORK

### A.1.1  FLOW MATCHING

Flow matching describes a class of deep generative models that approximate a time-dependent vector field $\boldsymbol{u}_t(\boldsymbol{x})$, $\boldsymbol{x} \in \mathbb{R}^d$ which generates a sequence of probability distributions $\{p_t : t \in [0,1]\}$ pushing a prior $p \equiv p_0$ towards the data distribution $q \equiv p_1$. The flow $\psi_t : [0,1] \times \mathbb{R}^d \rightarrow \mathbb{R}^d$ is defined through an ordinary differential equation (ODE) given the vector field $\boldsymbol{u}_t$:

$$\frac{d}{dt}\psi_t(\boldsymbol{x}) = \boldsymbol{u}_t(\psi_t(\boldsymbol{x})). \tag{6}$$

Efficient training of flow matching models is only possible because one does not need to define the true vector field $\boldsymbol{u}_t(\boldsymbol{x})$ but can instead match the conditional flow $\boldsymbol{u}_t(\boldsymbol{x}|\boldsymbol{x}_1)$ which is much easier to parameterize (Lipman et al., 2022) based on a data point $\boldsymbol{x}_1$. Thus, the conditional flow matching loss amounts to

$$\mathcal{L}_{\text{CFM}}(\theta) = \mathbb{E}_{t,q(\boldsymbol{x}_1),p(\boldsymbol{x}_0)}\|\boldsymbol{v}_\theta(\boldsymbol{x}_t,t) - \dot{\boldsymbol{x}}_t\|^2, \tag{7}$$

where $\dot{\boldsymbol{x}}_t = {}^{d}\!/\!{}_{dt}\,\psi_t(\boldsymbol{x}_0|\boldsymbol{x}_1)$ is the time derivative of the conditional flow $\boldsymbol{x}_t = \psi_t(\boldsymbol{x}_0|\boldsymbol{x}_1)$.

For sampling, we obtain a sample $\boldsymbol{x}_0$ from the prior $p$ and simulate the ODE in Eq. 6 replacing the true vector field $\boldsymbol{u}_t$ with the learned vector field $\boldsymbol{v}_\theta(\boldsymbol{x}_t, t)$.

In this work, we build on a variant called Independent-coupling Conditional Flow Matching (ICFM) (Albergo & Vanden-Eijnden, 2022; Tong et al., 2023) and consider a Gaussian conditional probability path

$$p_t(\boldsymbol{x}|\boldsymbol{x}_1) = \mathcal{N}(\boldsymbol{x}|\boldsymbol{\mu}_t(\boldsymbol{x}_1), \sigma_t(\boldsymbol{x}_1)^2\boldsymbol{I}) \tag{8}$$

with generating vector field

$$\boldsymbol{u}_t(\boldsymbol{x}|\boldsymbol{x}_1) = \frac{\sigma_t'(\boldsymbol{x}_1)}{\sigma_t(\boldsymbol{x}_1)}\left(\boldsymbol{x} - \boldsymbol{\mu}_t(\boldsymbol{x}_1)\right) + \boldsymbol{\mu}_t'(\boldsymbol{x}_1). \tag{9}$$

We use this setup with

$$\boldsymbol{\mu}_t(\boldsymbol{x}_1) = t\boldsymbol{x}_1 + (1 - t)\boldsymbol{x}_0, \quad \sigma_t(\boldsymbol{x}_1) = \sigma \tag{10}$$

to model the flow for ligand coordinates. This results in a constant velocity vector field

$$\dot{\boldsymbol{x}}_t = \frac{\boldsymbol{x}_1 - \boldsymbol{x}_t}{1 - t} = \boldsymbol{x}_1 - \boldsymbol{x}_0, \tag{11}$$

and the following flow matching loss:

$$\mathcal{L}_{\text{coord}}(\theta) = \mathbb{E}_{t,q(\boldsymbol{x}_1),p(\boldsymbol{x}_0)}\|\boldsymbol{v}_\theta(\boldsymbol{x}_t,t) - (\boldsymbol{x}_1 - \boldsymbol{x}_0)\|^2. \tag{12}$$

### A.1.2  RIEMANNIAN CONDITIONAL FLOW MATCHING

For side chain torsion angles, we need to define a flow on the torus $[-\pi,\pi)^N$. Fortunately, all components of the flow matching framework can be computed in a simulation-free manner on this simple manifold. We use the explicit Riemannian conditional flow matching (RCFM) loss derived by Chen & Lipman (2023):

$$\mathcal{L}_{\text{RCFM}}(\theta) = \mathbb{E}_{t,q(\boldsymbol{x}_1),p(\boldsymbol{x}_0)}\|\boldsymbol{v}_\theta(\boldsymbol{x}_t,t) - \dot{\boldsymbol{x}}_t\|_g^2. \tag{13}$$

The norm $\|\cdot\|_g$ on the tangent space $T_x\mathcal{M}$ at point $x$ on manifold $\mathcal{M}$ is induced by the Riemannian metric $g$ which is the standard inner product $\langle\boldsymbol{u},\boldsymbol{v}\rangle_g = \langle\boldsymbol{u},\boldsymbol{v}\rangle$, $\boldsymbol{u},\boldsymbol{v} \in T_x\mathcal{M}$ in this particular case.

Choosing the geodesic path to define the conditional flow $\boldsymbol{x}_t = \psi_t(\boldsymbol{x}_0|\boldsymbol{x}_1)$ we can compute intermediate points in closed-form as

$$\boldsymbol{x}_t = \exp_{\boldsymbol{x}_0}\left((1 - \kappa(t))\log_{\boldsymbol{x}_0}(\boldsymbol{x}_1)\right) \tag{14}$$

using the exponential and logarithm maps $\exp_{\boldsymbol{x}}(\boldsymbol{u}) = w(\boldsymbol{x} + \boldsymbol{u})$ and $\log_{\boldsymbol{x}}(\boldsymbol{y}) = \text{atan2}(\sin(\boldsymbol{y} - \boldsymbol{x}), \cos(\boldsymbol{y} - \boldsymbol{x}))$ [1]. $w(\alpha) = ((\alpha + \pi) \mod 2\pi) - \pi$ wraps values within the range $[-\pi,\pi)$ and is applied element-wise.

---

[1] $\boldsymbol{x}, \boldsymbol{y} \in \mathcal{M}$ and $\boldsymbol{u}, \boldsymbol{v} \in T_x\mathcal{M}$.

Here, we additionally use a scheduler $\kappa(t)$, that satisfies $\kappa(0) = 1$ and $\kappa(1) = 0$, to control the rate at which the geodesic distance $d$ between $\boldsymbol{x}_0$ and $\boldsymbol{x}_1$ decreases (Chen & Lipman, 2023):

$$d(\boldsymbol{x}_t, \boldsymbol{x}_1) = \kappa(t)d(\boldsymbol{x}_0, \boldsymbol{x}_1). \tag{15}$$

Thus, we obtain the loss function

$$\mathcal{L}_\chi(\theta) = \mathbb{E}_{t,q(\boldsymbol{x}_1),p(\boldsymbol{x}_0)}\|\boldsymbol{v}_\theta(\boldsymbol{x}_t, t) - \dot{\boldsymbol{x}}_t\|^2 \tag{16}$$

with

$$\dot{\boldsymbol{x}}_t = -\dot{\kappa}(t)\log_{\boldsymbol{x}_0}(\boldsymbol{x}_1). \tag{17}$$

Here, the learned vector field $\boldsymbol{v}_\theta(\boldsymbol{x}_t, t)$ represents vectors on the tangent plane.

### A.1.3 MARKOV BRIDGE MODEL

The molecular graph consists of discrete entities (node and edge types) and can therefore not be easily modeled in the flow matching framework. While discrete diffusion formulations (Austin et al., 2021; Vignac et al., 2022) can be used in principle, we decided to employ the Markov bridge model (Igashov et al., 2023) instead which is conceptually more similar to the flow matching scheme used for the continuous variables as it does not require a closed-form prior.

The Markov bridge model captures the stochastic dependency between two discrete-valued spaces $\mathcal{X}$ and $\mathcal{Y}$. It defines a Markov process between fixed start and end points $\boldsymbol{z}_0 = \boldsymbol{x}$ and $\boldsymbol{z}_1 = \boldsymbol{y}$, respectively, through a sequence of $N + 1$ random variables $(\boldsymbol{z}_{t=i/N})_{i=0}^N$ for which

$$p(\boldsymbol{z}_t|\boldsymbol{z}_0, \boldsymbol{z}_{0+\Delta t}, ..., \boldsymbol{z}_{t-\Delta t}, \boldsymbol{z}_1 = \boldsymbol{y}) = p(\boldsymbol{z}_t|\boldsymbol{z}_{t-\Delta t}, \boldsymbol{z}_1 = \boldsymbol{y}) \tag{18}$$

with $\Delta t = 1/N$. Additionally, since the process is pinned at its end point, we have

$$p(\boldsymbol{z}_1 = \boldsymbol{y}|\boldsymbol{z}_{1-\Delta t}, \boldsymbol{y}) = 1. \tag{19}$$

Each transition is given by

$$p(\boldsymbol{z}_{t+\Delta t}|\boldsymbol{z}_t, \boldsymbol{z}_1 = \boldsymbol{y}) = \text{Cat}(\boldsymbol{z}_{t+\Delta t}; \boldsymbol{Q}_t \boldsymbol{z}_t) \tag{20}$$

where $\boldsymbol{z}_t \in \{0, 1\}^K$ is a one-hot representation of the current category and $\boldsymbol{Q}_t$ is a transition matrix parameterised as

$$\boldsymbol{Q}_t \coloneqq \boldsymbol{Q}_t(\boldsymbol{y}) = \beta_t \boldsymbol{I} + (1 - \beta_t)\boldsymbol{y}\mathbf{1}_K^T. \tag{21}$$

Any intermediate state of the Markov chain can be probed in closed form:

$$p(\boldsymbol{z}_t|\boldsymbol{z}_0, \boldsymbol{z}_1) = \text{Cat}(\boldsymbol{z}_t; \bar{\boldsymbol{Q}}_{t-\Delta t}\boldsymbol{z}_0) \tag{22}$$

with

$$\bar{\boldsymbol{Q}}_t = \boldsymbol{Q}_t \boldsymbol{Q}_{t-\Delta t}...\boldsymbol{Q}_0 = \bar{\beta}_t \boldsymbol{I} + (1 - \bar{\beta}_t)\boldsymbol{y}\mathbf{1}_K^T. \tag{23}$$

In this work, we choose a linear schedule for $\bar{\beta} = 1 - t$ which implies $\beta_t = \bar{\beta}_t/\bar{\beta}_{t-\Delta t} = (1 - t)/(1 - t + \Delta t)$.

The neural network $\theta$ approximates $\boldsymbol{y}$ so that we can sample from the Markov bridge without knowing the true final state. It is trained by maximizing the following lower bound on the log-likelihood $q_\theta$ of the end point $\boldsymbol{y}$ given the start point $\boldsymbol{x}$

$$\log q_\theta(\boldsymbol{y}|\boldsymbol{x}) \geq -T \cdot \mathbb{E}_{t,\boldsymbol{z}_t \sim p(\boldsymbol{z}_t|\boldsymbol{x},\boldsymbol{y})} D_{\text{KL}}(p(\boldsymbol{z}_{t+\Delta t}|\boldsymbol{z}_t, \boldsymbol{y})\|q_\theta(\boldsymbol{z}_{t+\Delta t}|\boldsymbol{z}_t)) =: -\mathcal{L}_{\text{MBM}}(\theta). \tag{24}$$

### A.1.4 TRAINING LOSS

Our overall loss function is a weighted sum of the previously introduced loss terms:

$$\mathcal{L} = \lambda_{\text{coord}}\mathcal{L}_{\text{coord}} + \lambda_\chi \mathcal{L}_\chi + \lambda_a \mathcal{L}_{\text{MBM, atom}} + \lambda_b \mathcal{L}_{\text{MBM, bond}}. \tag{25}$$

A.2 PREDICTIVE UNCERTAINTY ESTIMATES FOR REGRESSION PROBLEMS

The basic idea behind our uncertainty estimation approach is to enable the neural network to not only output a single predicted value $\hat{y}$ but approximate the distribution within which the true value $y$ is likely to be found. Assuming this distribution is Gaussian, it can be specified with two parameters, mean and variance. To this end, we view the task of approximating the target variable $y \in \mathbb{R}^d$ as maximum likelihood estimation with a Gaussian model for the error $\epsilon = y - \hat{y}$ and maximise the probability density of the data under this Gaussian uncertainty model with $\hat{\Sigma} = \hat{\sigma}^2 I$:

$$p(y; \hat{y}, \hat{\Sigma}) = \mathcal{N}(y; \hat{y}, \hat{\sigma}^2 I) \tag{26}$$

$$= \frac{1}{\sqrt{(2\pi)^d \det \hat{\Sigma}}} \exp(-\frac{1}{2}(y - \hat{y})^T \hat{\Sigma}^{-1}(y - \hat{y})) \tag{27}$$

$$= \frac{1}{(2\pi)^{d/2} \hat{\sigma}^d} \exp(-\frac{1}{2\hat{\sigma}^2} \|y - \hat{y}\|^2) \tag{28}$$

where a neural network approximates both $\hat{y} := \hat{y}_\theta(x)$ and $\hat{\sigma} := \hat{\sigma}_\theta(x)$ based on the inputs $x$.

Maximising the log-likelihood yields

$$\arg \max_\theta [\log p(y; \hat{y}, \hat{\Sigma})] = \arg \max_\theta [-\frac{d}{2} \log 2\pi - d \log \hat{\sigma} - \frac{1}{2\hat{\sigma}^2} \|y - \hat{y}\|^2] \tag{29}$$

$$= \arg \max_\theta [-\frac{d}{2} \log \hat{\sigma}^2 - \frac{1}{2\hat{\sigma}^2} \|y - \hat{y}\|^2] \tag{30}$$

$$= \arg \min_\theta [\frac{d}{2} \log \hat{\sigma}^2 + \frac{1}{2\hat{\sigma}^2} \|y - \hat{y}\|^2], \tag{31}$$

which motivates the loss function

$$\mathcal{L}(\theta) = \frac{d}{2} \log \hat{\sigma}^2 + \frac{1}{2\hat{\sigma}^2} \|y - \hat{y}\|^2. \tag{32}$$

Intuitively, if the neural network encounters a rare training sample, it might be uncertain about its prediction and anticipate a large error $\|y - \hat{y}\|^2$. To keep the overall loss small regardless, it can now assign a larger variance $\hat{\sigma}^2$ to this "out-of-distribution"[2] sample. This means, according to our model, the true value might be found anywhere within a Gaussian distribution with a large spread. Moreover, the additive term $\frac{d}{2} \log \hat{\sigma}^2$ ensures the network does not achieve small loss values simply by excessively increasing $\hat{\sigma}^2$.

**Regularization** In Eq. 32, the standard regression loss is recovered if $\hat{\sigma}^2 = 1$. We find empirically that we can stabilize training through L2 regularization of the newly introduced parameter $\hat{\sigma}^2$ as follows:

$$\mathcal{L}(\theta) = \underbrace{\frac{d}{2} \log \hat{\sigma}^2 + \frac{1}{2\hat{\sigma}^2} \|y - \hat{y}\|^2}_{\substack{\text{Regression loss with} \\ \text{uncertainty estimation}}} + \underbrace{\frac{\lambda}{2} |\hat{\sigma}^2 - 1|^2}_{\substack{\text{Regularization of the} \\ \text{uncertainty estimate}}}. \tag{33}$$

This approach can be justified probabilistically by imposing a Gaussian prior with mean 1 and variance $1/\lambda$ on $\hat{\sigma}^2$. First, for simpler notation we redefine $p(y; \hat{y}, \hat{\Sigma})$ as $p(y|\hat{\sigma}^2) := p(y; \hat{y}, \hat{\Sigma})$. Next, using Bayes's theorem, we can convert the likelihood of an observed data point $y$ given the parameter $\hat{\sigma}^2$ into a posterior distribution:

$$p(\hat{\sigma}^2|y) \propto p(y|\hat{\sigma}^2) p(\hat{\sigma}^2). \tag{34}$$

---

[2]Note that we use the term "out-of-distribution" interchangeably with "uncertain" in the context of our generative model. Higher uncertainty implies that the sample is more likely to contain patterns dissimilar to the main modes of the data distribution as empirically supported by the data in Figure 2.

With prior $p(\hat{\sigma}^2) = \mathcal{N}(1, 1/\lambda)$, we can then solve the maximum *a posteriori* (MAP) problem as

$$\hat{\sigma}^2_{\text{MAP}} = \arg\max_{\hat{\sigma}^2}[\log p(\hat{\sigma}^2|\boldsymbol{y})] \tag{35}$$

$$= \arg\max_{\hat{\sigma}^2}[\log p(\boldsymbol{y}|\hat{\sigma}^2) + \log p(\hat{\sigma}^2)] \tag{36}$$

$$= \arg\min_{\hat{\sigma}^2}[\frac{d}{2}\log\hat{\sigma}^2 + \frac{1}{2\hat{\sigma}^2}\|\boldsymbol{y} - \hat{\boldsymbol{y}}\|^2 + \frac{(\hat{\sigma}^2 - 1)^2}{2(1/\lambda)}] \tag{37}$$

$$= \arg\min_{\hat{\sigma}^2}[\frac{d}{2}\log\hat{\sigma}^2 + \frac{1}{2\hat{\sigma}^2}\|\boldsymbol{y} - \hat{\boldsymbol{y}}\|^2 + \frac{\lambda}{2}|\hat{\sigma}^2 - 1|^2]. \tag{38}$$

Applying this idea to the flow matching setup, we obtain the loss in Eq. 1.

## A.3 MOTIVATION FOR THE FINAL UNCERTAINTY SCORE

In the flow matching setting, we integrate an ordinary differential equation (ODE) along a time-dependent vector field. However, the true data generating vector field is not known and we only have access to the estimated neural vector field $\boldsymbol{v}_\theta(\boldsymbol{x}_t, t)$ which introduces epistemic uncertainty into the process. We can model the true vector field $\boldsymbol{u}_t(\boldsymbol{x})$ as a sum of the estimated vector field and a normally distributed error term $\boldsymbol{\epsilon}_t \sim \mathcal{N}(\boldsymbol{0}, \sigma^2_\theta(\boldsymbol{x}_t, t)\boldsymbol{I})$

$$\boldsymbol{u}_t(\boldsymbol{x}) = \boldsymbol{v}_\theta(\boldsymbol{x}_t, t) + \boldsymbol{\epsilon}_t. \tag{39}$$

The newly introduced stochasticity can be described with a stochastic differential equation (SDE) rather than the original ODE

$$d\boldsymbol{x}_t = \boldsymbol{v}_\theta(\boldsymbol{x}_t, t)dt + \sigma_\theta(\boldsymbol{x}_t, t)d\boldsymbol{B}_t, \tag{40}$$

where $\boldsymbol{B}_t$ is the Wiener process.

As a result of integration, the final data point is

$$\boldsymbol{x}_t = \boldsymbol{x}_0 + \int_0^t \boldsymbol{v}_\theta(\boldsymbol{x}_s, s)ds + \int_0^t \sigma_\theta(\boldsymbol{x}_s, s)d\boldsymbol{B}_s. \tag{41}$$

We could find an approximate numerical solution using the Euler-Maruyama method:

$$\boldsymbol{x}_{t+\Delta t} = \boldsymbol{x}_t + \boldsymbol{v}_\theta(\boldsymbol{x}_t, t)\Delta t + \sigma_\theta(\boldsymbol{x}_t, t)\sqrt{\Delta t}\boldsymbol{z}, \tag{42}$$

where $\boldsymbol{z} \sim \mathcal{N}(\boldsymbol{0}, \boldsymbol{I})$. However, here we opt to use the learned vector field in the standard way and integrate it in a deterministic manner. This means we only have access to $\boldsymbol{v}_\theta(\cdot, s)$ and $\sigma_\theta(\cdot, s)$ along a deterministic trajectory. If we assume an alternative SDE parameterized by these values, we can write

$$\tilde{\boldsymbol{x}}_t = \boldsymbol{x}_0 + \int_0^t \boldsymbol{v}_\theta(s)ds + \int_0^t \sigma_\theta(s)d\boldsymbol{B}_s \tag{43}$$

and compute mean and variance of the final variable as[3]

$$\mathbb{E}[\tilde{\boldsymbol{x}}_t] = \boldsymbol{x}_0 + \int_0^t \boldsymbol{v}_\theta(s)ds, \tag{44}$$

$$\text{Var}[\tilde{\boldsymbol{x}}_t] = \int_0^t \sigma^2_\theta(s)ds. \tag{45}$$

This means we obtain the most likely data point according to our learned model by following the predicted mean as in conventional flow matching. However, this view point additionally provides us with an estimated variance of the sample. Note that Eq. 43 represents a purely hypothetical scenario created to motivate our final uncertainty score. More specifically, it does not describe actual DrugFlow sampling trajectories, which are fully deterministic except for the prior. Therefore $\mathbb{E}[\tilde{\boldsymbol{x}}_t]$ and $\text{Var}[\tilde{\boldsymbol{x}}_t]$ do not represent the mean and standard deviation of sampling outputs.

---

[3]Note that we cannot derive this formal solution if the stochastic process appears on both sides as in Eq. 41 because the integrands will contain expectations and variances as well.

## A.4 PREFERENCE ALIGNMENT

Our preference alignment scheme is based on Direct Preference Optimisation (DPO), a technique proposed by Rafailov et al. (2023) as a more stable alternative to reinforcement learning from human feedback methods aimed to optimise a model with respect to a target metric (i.e. reward function). Instead of explicitly learning the reward, Rafailov et al. (2023) proposed to fine-tune a pre-trained (*reference*) model $\theta$ on a synthetic dataset $\mathcal{D} = \{(x_i^w, x_i^l)\}_i$ collected from its own samples. This dataset consists of pairs of *winning* and *losing* samples $x_i^w$ and $x_i^l$, respectively, which are labeled according to problem-specific *preferences* (i.e. using human feedback or some other oracle). To align the method with these preferences, the authors introduce a new (*aligned*) model $\varphi$, initialised with $\theta$ and further optimised with a new loss that accounts for the provided preferences (Rafailov et al., 2023).

Initially proposed for aligning large language models, DPO was further adapted to diffusion models by Wallace et al. (2024). For noisy versions $x_t^w$ and $x_t^l$ (for clarity, we omit indices $i$) of the winning and losing data points, the diffusion DPO loss is computed using the true transition kernel $p$ and approximated transition kernels $q_\theta$ and $q_\varphi$ of the reference and aligned diffusion models as follows,

$$
\begin{aligned}
\mathcal{L}_{\text{DPO-Diffusion}}(\varphi) = -\mathbb{E}_{(x_1^w, x_1^l) \sim \mathcal{D}, t \sim \mathcal{U}(0,1), x_t^w \sim p(x_t^w|x_1^w), x_t^l \sim p(x_t^l|x_1^l)} \\
\log \sigma(-\beta T( \\
+ D_{\text{KL}}(p(x_{t+\Delta t}^w|x_1^w, t) \| q_\varphi(x_{t+\Delta t}^w|x_t^w)) \\
- D_{\text{KL}}(p(x_{t+\Delta t}^w|x_1^w, t) \| q_\theta(x_{t+\Delta t}^w|x_t^w)) \\
- D_{\text{KL}}(p(x_{t+\Delta t}^l|x_1^l, t) \| q_\varphi(x_{t+\Delta t}^l|x_t^l)) \\
+ D_{\text{KL}}(p(x_{t+\Delta t}^l|x_1^l, t) \| q_\theta(x_{t+\Delta t}^l|x_t^l)))).
\end{aligned}
\tag{46}
$$

We propose to apply this framework to the Markov bridge models. Using Eq. 24, we can derive:

$$
\mathcal{L}_{\text{DPO-MBM}}(\varphi) = -\log \sigma\left( -\beta(\mathcal{L}_{\text{MBM}}^w(\varphi) - \mathcal{L}_{\text{MBM}}^w(\theta) - \mathcal{L}_{\text{MBM}}^l(\varphi) + \mathcal{L}_{\text{MBM}}^l(\theta))\right).
\tag{47}
$$

Here, $\mathcal{L}_{\text{MBM}}^w(\varphi)$ and $\mathcal{L}_{\text{MBM}}^l(\varphi)$ are Markov bridge loss terms of the aligned model for winning and losing samples, respectively, and $\mathcal{L}_{\text{MBM}}^w(\theta)$ and $\mathcal{L}_{\text{MBM}}^l(\theta)$ are the loss terms for the pre-trained reference model $\theta$ with fixed parameters.

We then apply the same scheme to the coordinate flow matching loss and define the multi-domain preference alignment (MDPA) loss as a weighted sum of the different loss components. Furthermore, we scale the weighting constant by the sampled time $t \in [0, 1]$, as proposed in Cheng et al. (2024):

$$
\tilde{\mathcal{L}}_{\text{MDPA}}(\varphi) = -\log \sigma\left( -\beta t(\lambda_{\text{coord}}\Delta_{\text{coord}} + \lambda_{\text{atom}}\Delta_{\text{atom}} + \lambda_{\text{bond}}\Delta_{\text{bond}})\right),
\tag{48}
$$

where

$$
\begin{aligned}
\Delta_{\text{coord}} &= \mathcal{L}_{\text{coord}}^w(\varphi) - \mathcal{L}_{\text{coord}}^w(\theta) - \mathcal{L}_{\text{coord}}^l(\varphi) + \mathcal{L}_{\text{coord}}^l(\theta), \\
\Delta_{\text{atom}} &= \mathcal{L}_{\text{MBM, atom}}^w(\varphi) - \mathcal{L}_{\text{MBM, atom}}^w(\theta) - \mathcal{L}_{\text{MBM, atom}}^l(\varphi) + \mathcal{L}_{\text{MBM, atom}}^l(\theta), \\
\Delta_{\text{bond}} &= \mathcal{L}_{\text{MBM, bond}}^w(\varphi) - \mathcal{L}_{\text{MBM, bond}}^w(\theta) - \mathcal{L}_{\text{MBM, bond}}^l(\varphi) + \mathcal{L}_{\text{MBM, bond}}^l(\theta).
\end{aligned}
$$

Next, we introduce an additional regularization term, which is the scaled original loss (Eq. 25) applied to both winning and losing samples. The overall loss function is defined as the weighted sum of $\tilde{\mathcal{L}}_{\text{MDPA}}$ and the regularization term:

$$
\mathcal{L}_{\text{MDPA}}(\varphi) = \lambda_{\text{MDPA}}\tilde{\mathcal{L}}_{\text{MDPA}}(\varphi) + \lambda_w \mathcal{L}^w(\varphi) + \lambda_l \mathcal{L}^l(\varphi).
\tag{49}
$$

Setting $\lambda_{\text{MDPA}} = \lambda_l = 0$ corresponds to simple fine-tuning of the model on the winning samples, which we use as a baseline in our experiments.

## A.5 MODEL ARCHITECTURE AND TRAINING

**Input graph definition** While the computational graph of the generated small molecule must necessarily be complete so that bond types can be generated freely, we improve the computational

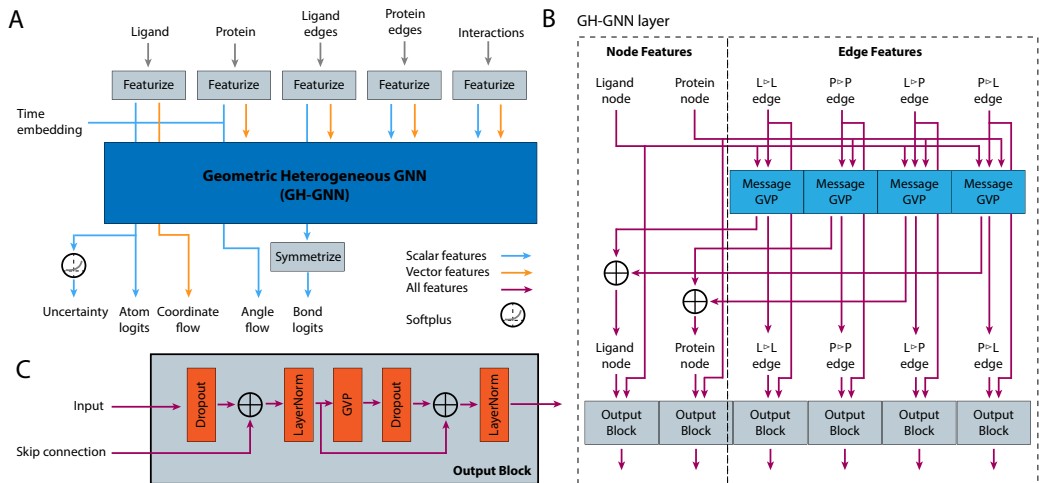

Figure 6: Architecture of our backbone neural network. (A) Different input types are featurized independently and processed with an $E(3)$-equivariant heterogeneous graph neural network based on Geometric Vector Perceptrons (GVP) (Jing et al., 2020). (B) One layer of the geometric heterogenous GNN (GH-GNN). Messages are computed based on source and destination node features and the corresponding edge features using GVPs. They are aggregated separately for appropriate destination node types and passed to the output block. (C) The output block of GH-GNN layers contains another equivariant GVP module.

efficiency by removing edges between pocket residues or between residues and ligand atoms based on a predefined cutoff distance (10Å). Nodes in this graph correspond either to a ligand atom or a pocket residue. The coordinates of the residue nodes are defined by the position of their $C_\alpha$ atoms. To retain the full atomic information while adopting this coarse-grained representation for the protein pocket (one computational node per residue), we include difference vectors to each atom of the residue in addition to the $C_\alpha$ coordinate and amino acid type as node input features similar to Zhang et al. (2023a). A schematic representation of the input graph showing different types of nodes and edges is provided in Figure 1.

**Featurization** We consider the atom types {C, N, O, S, B, Br, Cl, P, I, F, NH, N+, O-} where +/- indicate charges and NH is a nitrogen atom with explicit hydrogen. In all other cases, hydrogens are assumed to be implicit following normal valence assumptions. Furthermore, DRUGFLOW generates single, double, triple, aromatic, and "None" as bond types. FLEXFLOW additionally outputs five torsion angles {$\chi_1, \chi_2, \chi_3, \chi_4, \chi_5$} for each residue. Since not all angles are present in every residue we mask predictions where appropriate. For ligand nodes we include node-level cycle counts up to size 5 following Vignac et al. (2022) and Igashov et al. (2023).

**Self-conditioning** Self-conditioning (Chen et al., 2022) is a sampling strategy in which the neural network takes its previous prediction as additional input during iterative sampling. Like previous works (Yim et al., 2023b; Stärk et al., 2023) we observe significant performance improvements using this technique.

**Neural network** Since our computational graph contains two distinct groups of nodes, ligand and residue, and four different kinds of edges, ligand-to-ligand (L⊳L), ligand-to-pocket (L⊳P), pocket-to-ligand (P⊳L) and pocket-to-pocket (P⊳P), we use a heterogeneous graph neural network architecture as depicted in Figure 6. It performs message passing operations using separate learnable message functions for each edge type, and separate update functions for each node type. All these functions are implemented with geometric vector perceptron (GVP) layers (Jing et al., 2020; 2021) to ensure equivariance to global roto-translations.

**Number of nodes** To choose a number of computational nodes during sampling, which represent an upper bound on the final number of atoms, we compute the categorical distribution $p(N|M)$

(histogram) of molecule sizes $N$ given the number of residues $M$ in the target pocket based on the training set, sample from it, and add $N_{\mathrm{max}}/2$ extra nodes to account for the expected number of virtual nodes.

**Hyperparameters**  Important model hyperparameters are summarized in Table 4.

Table 4: Model hyperparameters.

| Parameter | Model | | | |
|---|---|---|---|---|
| | DRUGFLOW | DRUGFLOW-OOD | FLEXFLOW | DRUGFLOW-PA |
| Num. weights (M) | 12.1 | 12.1 | 12.6 | 12.1 |
| Model size (MB) | 48.408 | 48.411 | 50.557 | 48.408 |
| Training epochs | 600 | 600 | 700 | $\geq$50 |
| Virtual nodes $N_{\mathrm{max}}$ | 10 | 10 | 10 | 10 |
| Sampling steps | 500 | 500 | 500 | 500 |
| Uncertainty head | No | Yes | No | No |
| Flexible side chains | No | No | Yes | No |
| OOD $\lambda$ | – | 10 | – | – |
| Scheduler $k$ | – | – | 3 | – |
| Preference alignment | | | | |
| $\beta$ | – | – | – | 100 |
| $\lambda_{\mathrm{coord}}$ | – | – | – | 1 |
| $\lambda_{\mathrm{atom}}$ | – | – | – | 0.5 |
| $\lambda_{\mathrm{bond}}$ | – | – | – | 0.5 |
| $\lambda_w$ | – | – | – | 1 |
| $\lambda_l$ | – | – | – | 0.2 |

## B    EXTENDED RESULTS

### B.1    EXTENDED DISTRIBUTION LEARNING METRICS

In this section, we provide visual comparisons of distributions of various molecular characteristics from Tables 1, 2, and 3. First, we apply PCA to the molecular embeddings computed by ChemNet, the neural network used to calculate the Fréchet ChemNet Distance (FCD) (Preuer et al., 2018). We compare distributions of the first two principal components in Figure 7. Next, we compare distributions of discrete data types (atom and covalent bonds) in Figure 8. Finally, we provide violinplots for continuous distributions of various geometric and chemical properties, binding efficiency scores and normalised interaction counts in Figures 9, 10, and 11. Note that we remove outliers (beyond the 1st and the 99th percentiles) for better visibility. We additionally report Jensen-Shannon divergence for various geometric and chemical quantities, following the methodology chosen in other works (Guan et al., 2023a). To do this, we compute histograms of the scores splitting them in 100 bins of equal sizes on ranges defined by the minimum and maximum values of the corresponding quantities in the training data. The results are provided in Table 5.

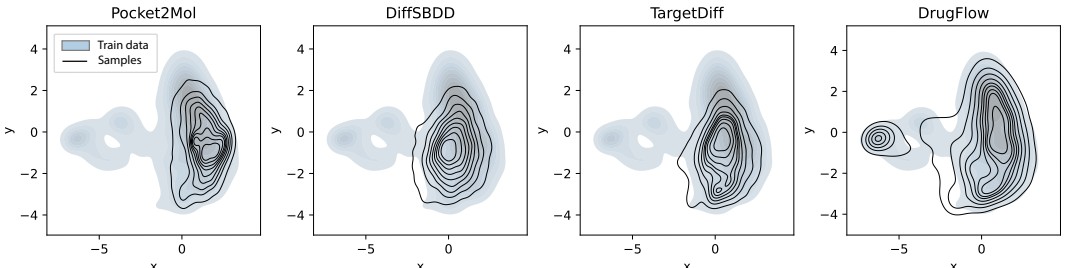

Figure 7: Distributions of the first two principle components of molecule embeddings computed by the FCD neural network. In each plot, we compare training data (blue areas) and samples generated by different methods (black solid lines).

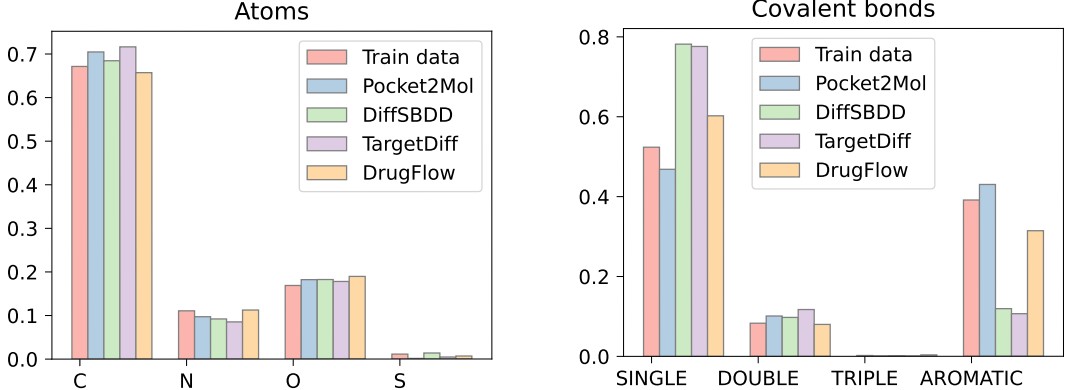

Figure 8: Distributions atom types (4 most popular types) and covalent bond types in the training data and samples.

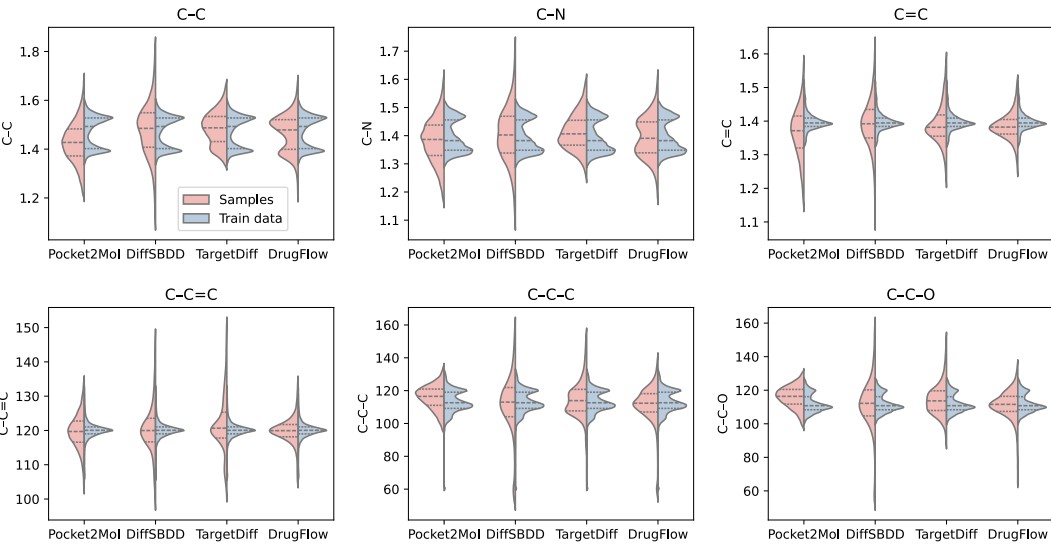

Figure 9: Distributions of bond distances and angles. Training data is visualized in blue and samples in red. Outliers falling beyond the 1st and the 99th percentiles are removed for better visibility.

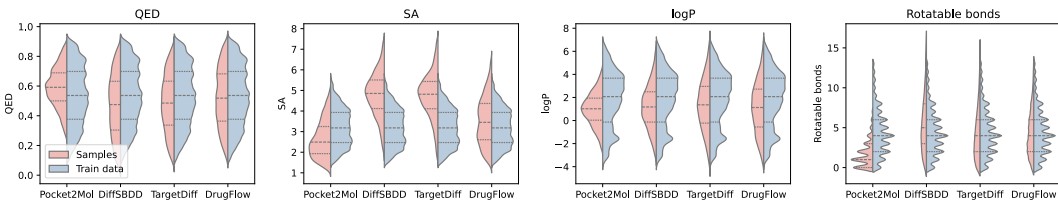

Figure 10: Distributions of molecular properties. Training data is visualized in blue and samples in red. Outliers falling beyond the 1st and the 99th percentiles are removed for better visibility.

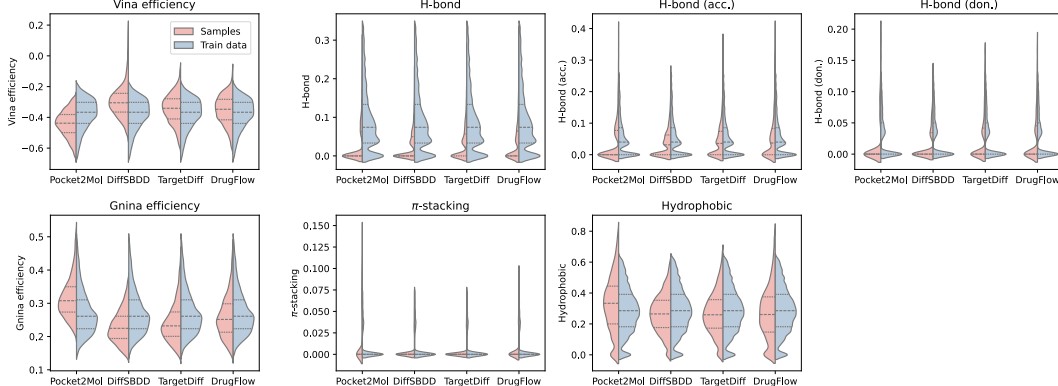

Figure 11: Distributions of binding efficiency scores and normalised numbers of interactions. Training data is visualized in blue and samples in red. Outliers falling beyond the 1st and the 99th percentiles are removed for better visibility.

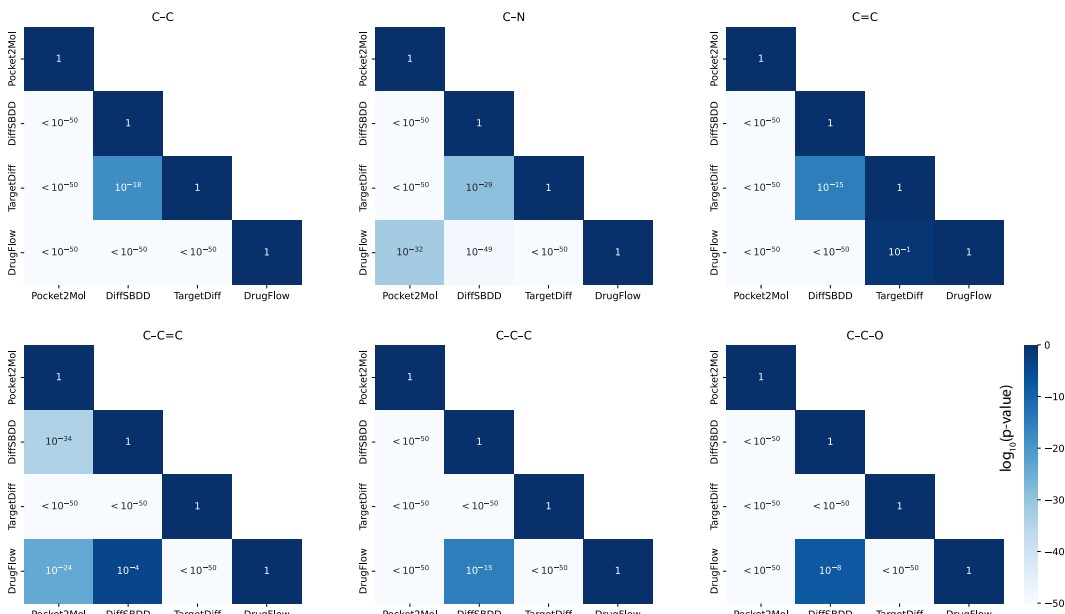

Figure 12: Mann-Whitney U test to assess whether there is a statistically significant difference between samples across various scores and methods. Here, we compare distributions of bond lengths and angles. We perform pairwise comparisons of distributions for all possible pairs of methods and report the resulting p-values. As shown in the last row of each matrix, the distribution differences between DRUGFLOW and other methods are statistically significant almost everywhere.

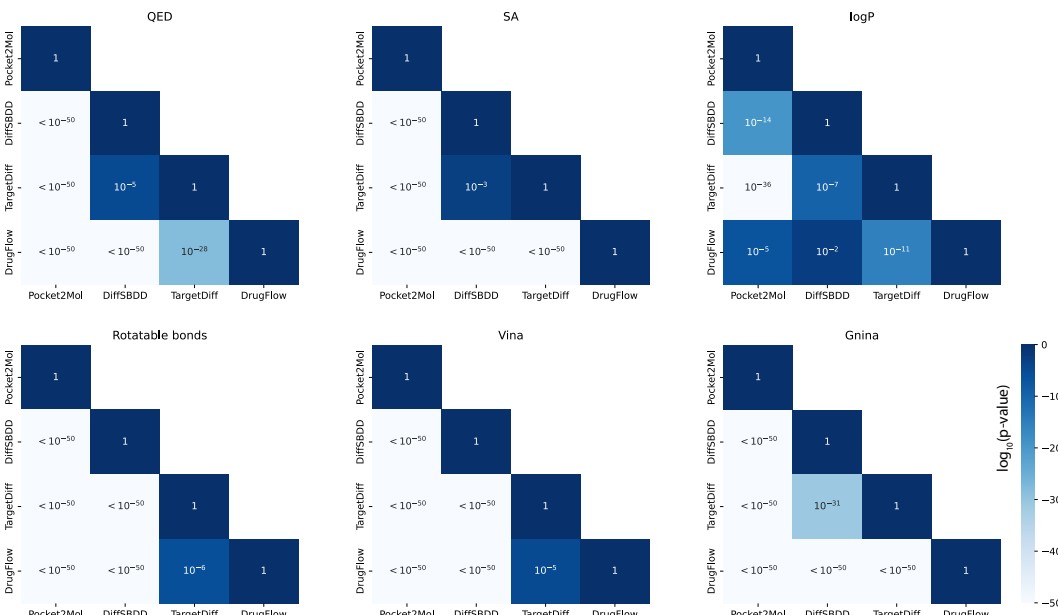

Figure 13: Mann-Whitney U test to assess whether there is a statistically significant difference between samples across various scores and methods. Here, we compare distributions of molecular properties and binding efficiency scores. We perform pairwise comparisons of distributions for all possible pairs of methods and report the resulting p-values. As shown in the last row of each matrix, the distribution differences between DRUGFLOW and other methods are statistically significant almost everywhere.

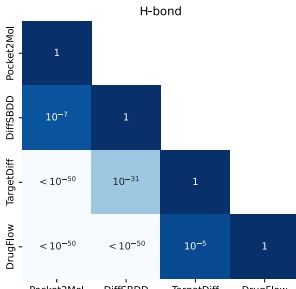 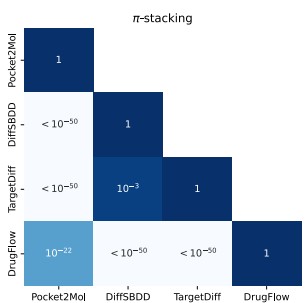 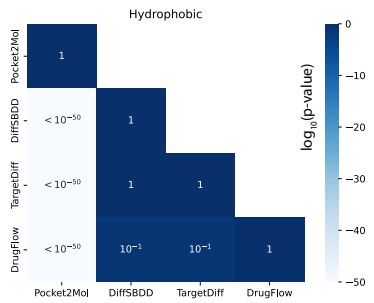

Figure 14: Mann-Whitney U test to assess whether there is a statistically significant difference between samples across various scores and methods. Here, we compare distributions of normalised numbers of protein-ligand interactions. We perform pairwise comparisons of distributions for all possible pairs of methods and report the resulting p-values. As shown in the last row of each matrix, the distribution differences between DRUGFLOW and other methods are statistically significant almost everywhere.

Table 5: Jensen-Shannon divergence between distributions of continuous molecular data (bond distances and angles), drug-likeness (QED), synthetic accessibility (SA), lipophilicity (logP) and numbers of rotatable bonds (RB). The best result is highlighted in bold, the second best is underlined.

| Method | Top-3 bond distances | | | Top-3 bond angles | | | Molecular properties | | | |
|---|---|---|---|---|---|---|---|---|---|---|
| | C–C | C–N | C=C | C–C=C | C–C–C | C–C–O | QED | SA | logP | Rotatable bonds |
| POCKET2MOL | 0.357 | 0.289 | 0.400 | _0.312_ | _0.197_ | 0.276 | 0.247 | _0.230_ | 0.330 | 0.400 |
| DIFFSBDD | 0.339 | 0.312 | 0.354 | 0.329 | 0.315 | 0.343 | 0.162 | 0.479 | 0.203 | 0.115 |
| TARGETDIFF | _0.236_ | _0.219_ | _0.321_ | 0.317 | 0.222 | _0.254_ | _0.138_ | 0.470 | **0.148** | _0.083_ |
| DRUGFLOW | **0.223** | **0.218** | **0.242** | **0.164** | **0.154** | **0.177** | **0.089** | **0.170** | _0.156_ | **0.064** |

## B.2 ABSOLUTE METRICS

In Table 6, we provide additional metrics evaluating overall quality of samples in absolute values. For reference, we also provide the training set numbers and remind that it is unreasonable to expect the model trained solely with the likelihood objective to substantially surpass metric values from the training data.

Table 6: Absolute values of various quality metrics. The best result is highlighted in bold, the second best is underlined.

| Method | Size | Passed filters, % | | | Binding efficiency | | Uniqueness ↑ | Novelty ↑ |
|---|---|---|---|---|---|---|---|---|
| | | PoseBusters ↑ | REOS ↑ | Clashes ↑ | Gnina ↑ | Vina ↓ | | |
| POCKET2MOL | 13.024 | **0.867** | **0.408** | **0.926** | **0.318** | **-0.443** | 0.892 | 0.996 |
| DIFFSBDD | 24.397 | 0.380 | 0.110 | 0.696 | 0.231 | -0.293 | **1.000** | **1.000** |
| TARGETDIFF | 24.150 | 0.510 | 0.174 | 0.884 | 0.244 | -0.345 | _0.989_ | _0.998_ |
| DRUGFLOW | 20.827 | _0.731_ | _0.245_ | _0.897_ | _0.261_ | _-0.351_ | 0.956 | 0.997 |
| Training set | 23.648 | 0.948 | 0.248 | 0.977 | 0.274 | -0.379 | — | — |

## B.3 ABLATION STUDIES

Here, we study the effect of using virtual nodes, uncertainty estimation and flexible side chains on the metrics we reported in the main text. We repeat the same evaluation procedure and compare four models:

- DRUGFLOW, our base model from Tables 1, 2, and 3. This model uses virtual nodes but not the uncertainty head;
- DRUGFLOW (no virt. nodes), an identical model without virtual nodes;
- DRUGFLOW-OOD, an identical model trained with additional uncertainty head to be able to detect out-of-distribution (OOD) samples. This model uses virtual nodes;
- FLEXFLOW, an identical model that additionally operates on protein side chains. This model uses virtual nodes but does not have the uncertainty head.

As shown in Tables 7, 8, and 9, DRUGFLOW-OOD demonstrates competitive performance with DRUGFLOW. Notably, it remarkably improves Wasserstein distance on bond angles, QED, and SA scores. The model without virtual nodes demonstrates consistently worse performance across all metrics except lipophilicity. FLEXFLOW performs worse on metrics related to pocket interactions. This result is expected due to the much higher complexity of the flexible design task. While DRUGFLOW learns the conditional distribution of molecules given fixed (ground-truth) pockets, FLEXFLOW learns the joint distribution of molecules and side chain conformations.

To further contextualize the performance of our models, we include a simple, unconditional baseline. For each test set protein, we randomly selected and docked 100 molecules from the 2.4M compounds in the ChEMBL database (release 34). We also repeated this procedure for CrossDocked training set molecules to provide another simple baseline.

The results in Tables 7 and 8 highlight the importance of training set curation for generating molecules with desirable properties. Most molecular properties (such as QED, SA score, and logP) deviate more from the CrossDocked training set than DrugFlow's generated molecules. The FCD is substantially higher as well. A notable exception are more fundamental molecular features like bond lengths and angles for which the ChEMBL baseline is competitive. This can be explained by the universal physical rules all molecules must obey. Furthermore, Table 9 demonstrates the importance of pocket-conditioning. Because ChEMBL molecules have been selected without taking into account the structure of the binding pocket, the distributions of most interaction-related features are matched consistently worse by this baseline than any of the DrugFlow variants.

Lastly, we also study the performance of DRUGFLOW depending on the number of training epochs. As shown in Tables 10 and 11, the performance varies with training and is generally improving (Validity, FCD, Rings, RB), as can be expected. Some other properties such as atom types, bond types and geometries (distances and angles) are however learned rather quickly, and more training alone does not seem to guarantee better results.

Table 7: Fréchet ChemNet Distance and Jensen-Shannon divergence between distributions of discrete molecular data. The best result is highlighted in bold, the second best is underlined.

| Method | FCD | Atoms | Bonds | Rings |
|---|---|---|---|---|
| DRUGFLOW | 4.278 | **0.043** | 0.060 | 0.391 |
| DRUGFLOW-OOD | 4.328 | 0.054 | 0.019 | 0.374 |
| DRUGFLOW (no virt. nodes) | 5.380 | 0.050 | 0.093 | 0.411 |
| FLEXFLOW | 6.001 | 0.086 | 0.068 | 0.376 |
| CHEMBL | 9.255 | 0.124 | 0.038 | 0.377 |
| CROSSDOCKED | **0.578** | 0.066 | **0.003** | **0.186** |

Table 8: Wasserstein distance between distributions of continuous molecular data (bond distances and angles), drug-likeness (QED), synthetic accessibility (SA), lipophilicity (logP) and numbers of rotatable bonds (RB). The best result is highlighted in bold, the second best is underlined.

| Method | Top-3 bond distances | | | Top-3 bond angles | | | Molecular properties | | | |
|---|---|---|---|---|---|---|---|---|---|---|
| | C–C | C–N | C=C | C–C=C | C–C–C | C–C–O | QED | SA | logP | RB |
| DRUGFLOW | 0.017 | 0.016 | 0.016 | 0.952 | 2.269 | 1.941 | 0.014 | 0.317 | 0.665 | 0.144 |
| DRUGFLOW-OOD | 0.021 | 0.029 | 0.021 | 0.683 | 1.412 | 1.478 | 0.005 | 0.140 | 0.796 | 0.176 |
| DRUGFLOW (no virt. nodes) | 0.027 | 0.028 | 0.018 | 1.149 | 2.681 | 1.964 | 0.021 | 0.512 | 0.658 | 0.214 |
| FLEXFLOW | 0.019 | 0.019 | 0.017 | 1.021 | 1.731 | 1.937 | 0.029 | 0.231 | 1.115 | 0.497 |
| CHEMBL | 0.024 | 0.022 | 0.020 | 0.536 | 0.517 | 2.966 | 0.028 | 0.322 | 1.656 | 1.456 |
| CROSSDOCKED | 0.001 | 0.001 | 0.000 | 0.013 | 0.085 | 0.060 | 0.002 | 0.011 | 0.028 | 0.041 |

Table 9: Wasserstein distance between distributions of binding efficiency scores and normalized numbers of different protein-ligand interactions. The best result is highlighted in bold, the second best is underlined.

| Method | Binding efficiency | | Protein-ligand interactions | | | | | |
|---|---|---|---|---|---|---|---|---|
| | Vina | Gnina | H-bond | H-bond (acc.) | H-bond (don.) | $\pi$-stacking | Hydrophobic |
| DRUGFLOW | 0.028 | 0.013 | 0.019 | 0.012 | 0.007 | 0.006 | 0.036 |
| DRUGFLOW-OOD | 0.044 | 0.019 | 0.027 | 0.015 | 0.011 | 0.003 | 0.043 |
| DRUGFLOW (no virt. nodes) | 0.054 | 0.020 | 0.030 | 0.018 | 0.011 | 0.007 | 0.038 |
| FLEXFLOW | 0.073 | 0.040 | 0.077 | 0.052 | 0.025 | 0.012 | 0.203 |
| CHEMBL | 0.176 | 0.078 | 0.083 | 0.055 | 0.028 | 0.003 | 0.061 |
| CROSSDOCKED | 0.149 | 0.071 | 0.061 | 0.042 | 0.019 | 0.004 | 0.081 |

## B.4 UNCERTAINTY ESTIMATION

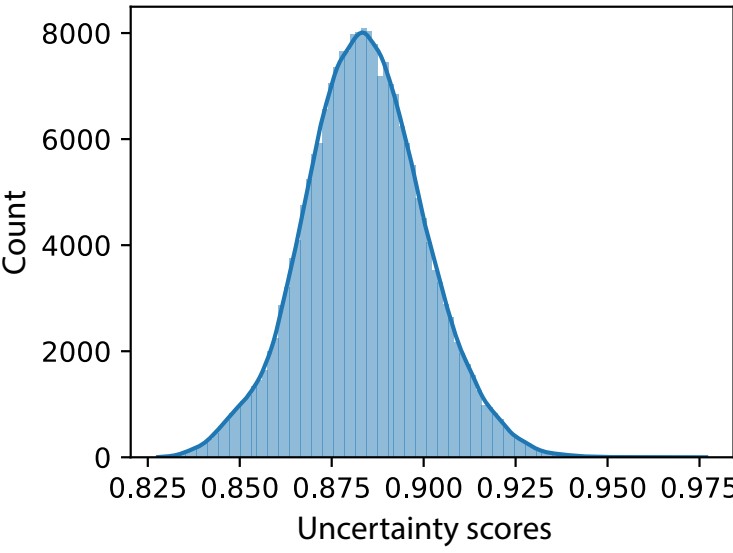

Figure 15: Distribution of the uncertainty scores on the test set.

Table 10: Dependency of the evaluation results on the number of training epochs. Validity, Fréchet ChemNet Distance and Jensen-Shannon divergence between distributions of discrete molecular data. The best result is highlighted in bold, the second best is underlined.

| Epochs | Validity | FCD | Atoms | Bonds | Rings |
|---|---|---|---|---|---|
| 100 | 0.785 | 6.064 | 0.044 | 0.070 | 0.476 |
| 300 | 0.807 | 5.775 | 0.078 | **0.045** | 0.416 |
| 500 | 0.885 | 5.154 | 0.067 | 0.063 | 0.416 |
| 600 | **0.891** | **4.278** | **0.043** | 0.060 | **0.391** |

Table 11: Dependency of the evaluation results on the number of training epochs. For all scores, Wasserstein distance between the corresponding distributions is reported. The best result is highlighted in bold, the second best is underlined. RB: number of rotatable bonds.

| | Top-3 bond distances | | | Top-3 bond angles | | | Molecular properties | | | |
|---|---|---|---|---|---|---|---|---|---|---|
| Epochs | C–C | C–N | C=C | C–C=C | C–C–C | C–C–O | QED | SA | logP | RB |
| 100 | 0.040 | 0.035 | 0.025 | 1.021 | 2.035 | 1.811 | 0.022 | 0.655 | **0.344** | 0.719 |
| 300 | **0.015** | 0.020 | **0.014** | **0.508** | 1.533 | **1.372** | 0.055 | 0.589 | 1.260 | 0.515 |
| 500 | 0.016 | 0.024 | 0.018 | 0.883 | **1.417** | 1.417 | 0.014 | 0.438 | 0.487 | 0.347 |
| 600 | 0.017 | **0.016** | 0.016 | 0.952 | 2.269 | 1.941 | **0.014** | **0.317** | 0.665 | **0.144** |

## B.5 VIRTUAL NODES

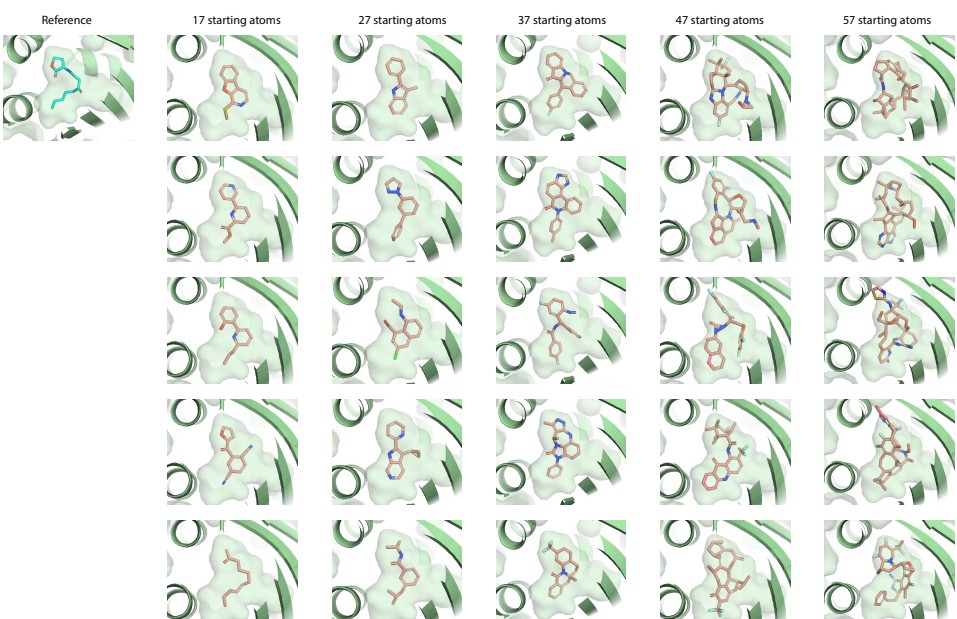

Figure 16: Samples for the pocket 1L3L with varied input size.

Table 12: Dependency of the evaluation results on the number of training epochs. Wasserstein distance between distributions of binding efficiency scores and normalized numbers of different protein-ligand interactions. The best result is highlighted in bold, the second best is underlined.

| Training epochs | Binding efficiency | | Protein-ligand interactions | | | | |
|---|---|---|---|---|---|---|---|
| | Vina | Gnina | H-bond | H-bond (acc.) | H-bond (don.) | $\pi$-stacking | Hydrophobic |
| 100 | 0.038 | 0.019 | 0.028 | 0.016 | 0.012 | 0.007 | **0.028** |
| 300 | 0.039 | 0.035 | **0.019** | **0.011** | 0.008 | **0.004** | 0.075 |
| 500 | 0.056 | 0.029 | 0.024 | 0.011 | 0.013 | 0.005 | 0.041 |
| 600 | **0.028** | **0.013** | 0.019 | 0.012 | **0.007** | 0.006 | 0.036 |

## B.6 DISTRIBUTION OF SIDE CHAIN ANGLES

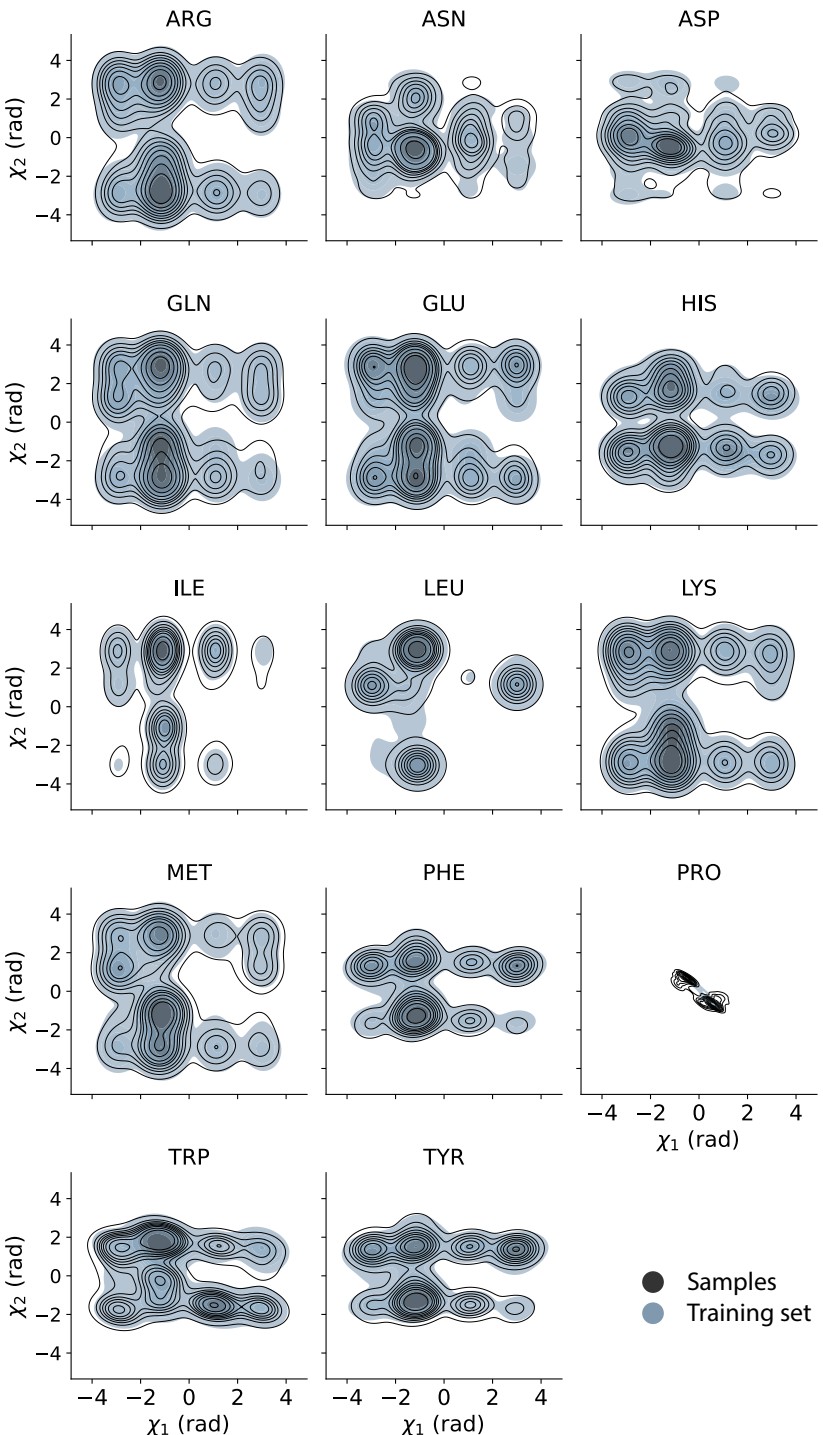

Figure 17: Distributions of $\chi_1$ and $\chi_2$ angles for the 14 amino acids that have at least two side chain torsion angles. We compare FLEXFLOW samples to the bound pocket conformations from the training set.

## B.7 Additional preference alignment results

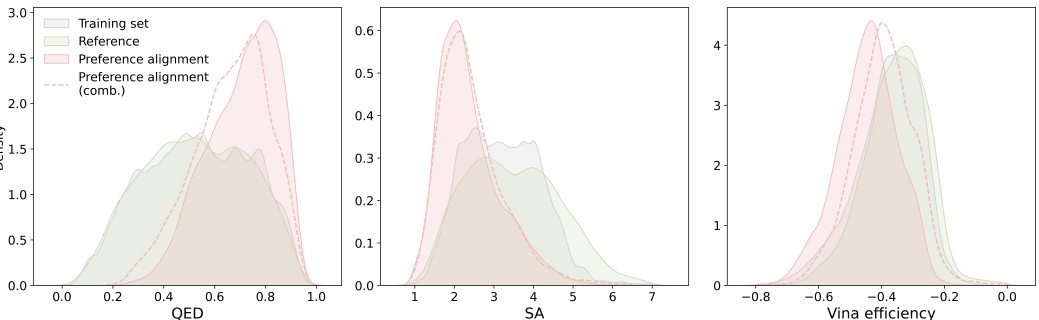

Figure 18: **Preference alignment shifts property distributions.** Distributions of QED, SA, and Vina efficiency values with and without preference alignment. Gray shaded areas represent the training set distributions, and green areas show distributions from DRUGFLOW as the reference model. Solid red lines indicate the distributions for preference-aligned models for each specific property, while the dashed red line shows the distribution for the model aligned with combined preferences. Preference alignment leads to significant shifts toward more desirable values in QED, SA, and Vina efficiency across all metrics.

Table 13: **Preference alignment comparison.** Performance comparison of preference-aligned (PA) models vs. fine-tuned (FT) models on REOS, QED, SA, Vina efficiency, and combined preference pairs, using DRUGFLOW as the baseline. We also report Vina scores (after local minimization), even though our models were not directly optimized for these scores. Values for AliDiff (Gu et al., 2024) and DecompDPO (Cheng et al., 2024) are as reported by their authors (SA values were mapped to the original scale using $SA = 10 - 9SA_{norm}$). The authors of AliDiff provide sampled molecules and Vina scores, which we use for re-evaluation.[4] Bold values indicate the best performance, and underlined values indicate the second-best. Molecular validity, uniqueness, novelty, and PoseBusters success rates are also reported. The preference-aligned models achieve the highest performance on the target metrics, with combined preference alignment models ranking second. Preference alignment models show a 10-20% drop in molecular validity compared to less than 5% for fine-tuned models. PoseBusters measures robustness against common failure modes of generative models (Buttenschoen et al., 2024).

| Method | Molecular properties | | | | | | | Interactions | |
| | Valid. ↑ | Uniq. ↑ | Nov. ↑ | PoseB. ↑ | REOS ↑ | QED ↑ | SA ↓ | Vina eff. ↓ | Vina min ↓ |
| --- | --- | --- | --- | --- | --- | --- | --- | --- | --- |
| Baseline | 0.89 | 0.95 | 1.00 | **0.73** | 0.25 | 0.53 | 3.49 | -0.35 | -6.67 |
| FT (REOS) | 0.87 | **1.00** | 1.00 | 0.62 | 0.55 | 0.57 | 3.85 | -0.29 | -7.35 |
| FT (QED) | 0.85 | **1.00** | 1.00 | 0.62 | 0.28 | 0.58 | 3.88 | -0.30 | -7.33 |
| FT (SA) | 0.86 | **1.00** | 1.00 | 0.58 | 0.24 | 0.50 | 3.45 | -0.30 | -7.41 |
| FT (Vina efficiency) | 0.77 | **1.00** | 1.00 | 0.52 | 0.22 | 0.50 | 4.37 | -0.33 | -8.25 |
| FT (Combined) | 0.84 | **1.00** | 1.00 | 0.57 | 0.51 | 0.58 | 3.74 | -0.30 | -7.47 |
| PA (REOS) | 0.79 | 0.97 | 1.00 | 0.67 | **0.88** | 0.65 | 3.90 | -0.37 | -7.08 |
| PA (QED) | 0.78 | 0.99 | 1.00 | 0.52 | 0.41 | **0.71** | 3.62 | -0.36 | -6.80 |
| PA (SA) | 0.76 | 0.93 | 1.00 | 0.45 | 0.29 | 0.57 | **2.39** | -0.35 | -6.50 |
| PA (Vina efficiency) | 0.65 | 0.95 | 1.00 | 0.37 | 0.29 | 0.56 | 3.83 | **-0.45** | -7.78 |
| PA (Combined) | 0.73 | 0.94 | 1.00 | 0.69 | 0.76 | 0.67 | 2.45 | -0.40 | -7.18 |
| AliDiff (our evaluation)[4] | **0.92** | 0.99 | 1.00 | ≤ 0.26 | 0.20 | 0.50 | 4.92 | -0.34 | -7.87 |
| AliDiff (Vina) | - | - | - | - | - | 0.50 | 4.87 | - | -8.09 |
| AliDiff (Vina, SA) | - | - | - | - | - | 0.52 | 4.60 | - | -8.00 |
| AliDiff (Vina, QED) | - | - | - | - | - | 0.51 | 4.87 | - | -8.01 |
| DecompDPO (Vina) | - | - | - | - | - | 0.48 | 4.06 | - | **-8.49** |
| DecompDPO (Vina, SA, QED) | - | - | - | - | - | 0.48 | 4.24 | - | -7.93 |
| Training set | 1.00 | - | - | 0.95 | 0.25 | 0.53 | 3.23 | -0.38 | -8.29 |

---

[4]For evaluating AliDiff, molecules sampled for the CrossDocked test set were retrieved from github.com/MinkaiXu/AliDiff. For PoseBusters, only pocket-independent checks were completed, as pocket information was unavailable in the provided samples. Vina efficiency scores were derived from the reported Vina minimization scores.

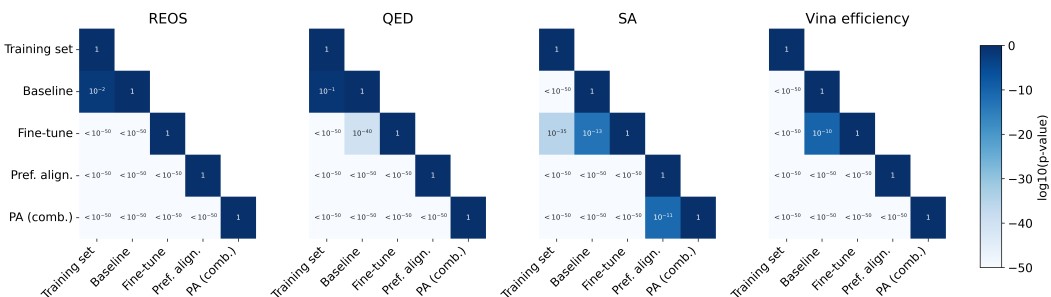

Figure 19: Mann-Whitney U test to assess whether there is a statistically significant difference between different preference alignment methods for REOS, QED, SA, and Vina efficiency scores reported in Figure 5. We perform pairwise comparisons of distributions for all possible pairs of methods and report the resulting p-values. As shown in the last two rows of each matrix, the distribution differences between the preference aligned model and other methods are statistically significant everywhere.

### B.8  PREDICTION ERROR

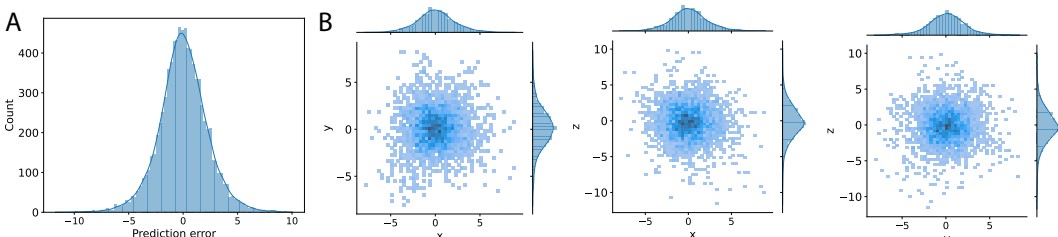

Figure 20: Distribution of the prediction error of a trained DRUGFLOW model. The values are computed as $v_\theta(x_t, t) - (x_1 - x_0)$ for one training batch. (A) All vector components are treated as independent (1D) samples. (B) Joint distributions for pairs of error components.

## B.9 Distribution of the joint QED, SA, logP and Vina scores

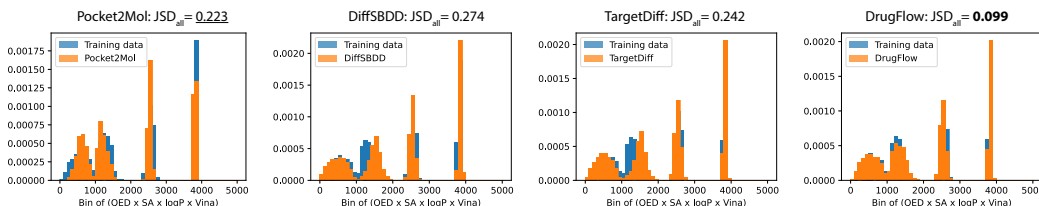

Figure 21: Histograms of the joint distributions of QED, SA, logP and Vina efficiency scores (10 bins per score). We show top-5000 bins as the rest is not populated.

## B.10 Significance of Differences in Wasserstein Distance and Jensen-Shannon Divergence

To evaluate the variability and statistical significance of sample-based Wasserstein distances and Jensen-Shannon divergences, we generated 20 bootstrap samples, each containing 500 data points (5 samples per test target). These samples were used to compute distances to the training set, as in Section 3.1. Tables 14–16 provide the mean values and standard deviations of these distances over the bootstrapped samples. Statistical significance of the DrugFlow's superior performance is verified with Student's t-test, as shown in Figures 22–24.

Table 14: Sample mean and standard deviation of the Jensen-Shannon divergence between distributions of molecular data, as presented in Table 3. The final column shows the Jensen-Shannon divergence for the joint distributions of four molecular properties: QED, SA, LogP, and Vina efficiency. Standard deviations are provided in brackets.

| Method | Atoms | Bonds | Rings | JSD$_{all}$ |
|---|---|---|---|---|
| POCKET2MOL | 0.082 (0.004) | **0.045** (0.006) | 0.491 (0.011) | 0.226 (0.011) |
| DIFFSBDD | 0.052 (0.003) | 0.227 (0.005) | 0.613 (0.007) | 0.277 (0.005) |
| TARGETDIFF | 0.078 (0.004) | 0.240 (0.011) | 0.649 (0.007) | 0.247 (0.010) |
| DRUGFLOW | **0.044** (0.005) | 0.060 (0.005) | **0.433** (0.006) | **0.106** (0.006) |

Table 15: Sample mean and standard deviation of Wasserstein distances reported in Table 1.

| | Top-3 bond distances | | | Top-3 bond angles | | | Molecular properties | | | |
|---|---|---|---|---|---|---|---|---|---|---|
| Method | C–C | C–N | C=C | C–C=C | C–C–C | C–C–O | QED | SA | logP | RB |
| POCKET2MOL | 0.050 (0.001) | 0.024 (0.001) | 0.046 (0.002) | 2.174 (0.123) | 2.967 (0.262) | 3.957 (0.254) | 0.073 (0.004) | 0.576 (0.057) | 1.210 (0.036) | 2.861 (0.071) |
| DIFFSBDD | 0.041 (0.001) | 0.039 (0.002) | 0.042 (0.002) | 3.632 (0.193) | 8.165 (0.250) | 7.754 (0.301) | 0.065 (0.008) | 1.571 (0.030) | 0.781 (0.047) | 0.935 (0.188) |
| TARGETDIFF | 0.018 (0.001) | 0.019 (0.001) | 0.028 (0.002) | 4.277 (0.235) | 3.430 (0.169) | 4.149 (0.240) | 0.053 (0.017) | 1.527 (0.082) | **0.514** (0.114) | 0.420 (0.101) |
| DRUGFLOW | **0.017** (0.001) | **0.017** (0.001) | **0.016** (0.002) | **0.957** (0.083) | **2.276** (0.336) | **1.972** (0.292) | **0.015** (0.005) | **0.320** (0.046) | 0.682 (0.056) | **0.214** (0.053) |

Table 16: Sample mean and standard deviation of Wasserstein distances reported in Table 2.

| | Binding efficiency | | Protein-ligand interactions | | | | |
|---|---|---|---|---|---|---|---|
| Method | Vina | Gnina | H-bond | H-bond (acc.) | H-bond (don.) | $\pi$-stacking | Hydrophobic |
| POCKET2MOL | 0.064 (0.005) | 0.044 (0.005) | 0.040 (0.003) | 0.026 (0.002) | 0.014 (0.001) | 0.008 (0.001) | **0.028** (0.008) |
| DIFFSBDD | 0.086 (0.004) | 0.043 (0.002) | 0.047 (0.002) | 0.030 (0.002) | 0.017 (0.002) | 0.011 (0.001) | 0.044 (0.003) |
| TARGETDIFF | 0.036 (0.007) | 0.031 (0.004) | 0.055 (0.012) | 0.045 (0.009) | 0.013 (0.005) | 0.012 (0.003) | 0.063 (0.024) |
| DRUGFLOW | **0.028** (0.005) | **0.013** (0.003) | **0.019** (0.003) | **0.012** (0.002) | **0.008** (0.001) | **0.006** (0.001) | 0.037 (0.004) |

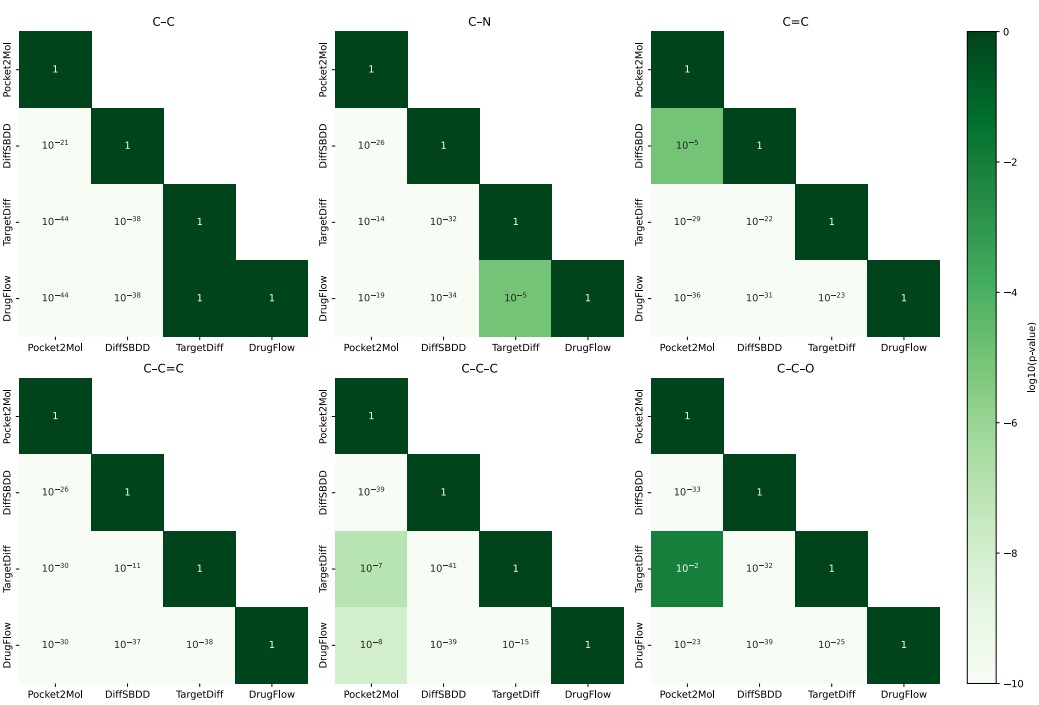

Figure 22: Statistical analysis of Wasserstein distance differences using Student's t-test across various metrics and methods for bond lengths and angles. Wasserstein distances were computed using $n = 20$ bootstrap samples, each containing $500$ datapoints. A two-sample t-test was conducted to determine whether the differences of the sample means of the Wasserstein distances were statistically significant. The p-values are reported on a log-scale and visualized as a heatmap. As illustrated in the last row of each matrix, the differences between DRUGFLOW and other methods are statistically significant in nearly all cases.

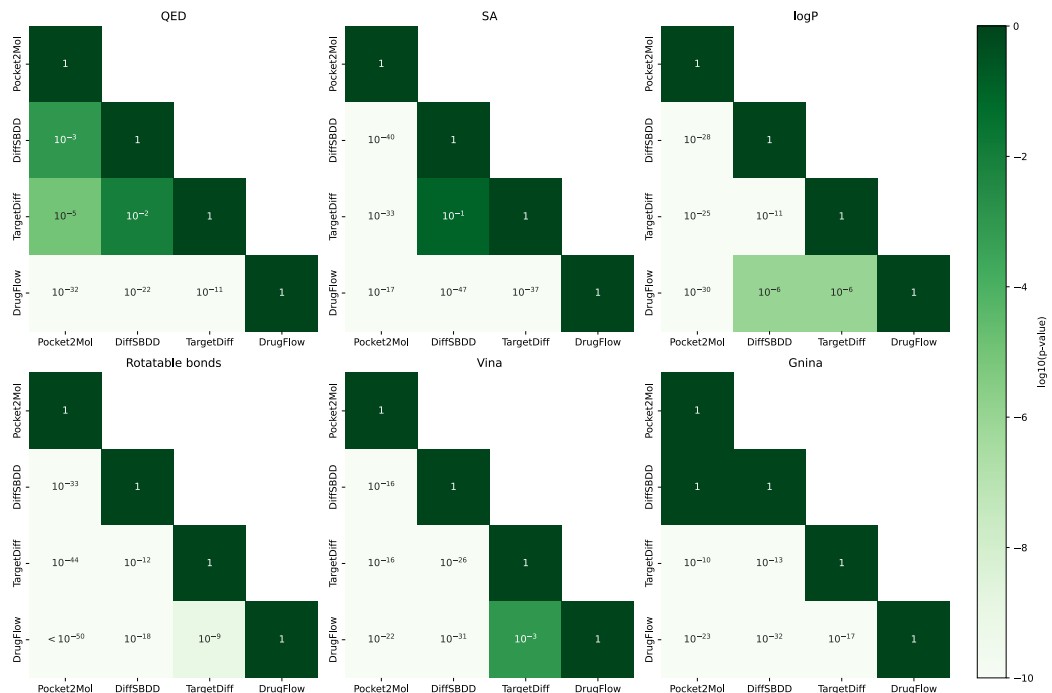

Figure 23: Statistical analysis of Wasserstein distance differences using Student's t-test across molecular property distributions, including QED, SA, LogP, number of rotatable bonds, and Vina/Gnina efficiency. As in Figure 22, p-values are displayed on a log-scale heatmap, with the last row indicating that distance differences for DRUGFLOW are statistically significant in all cases.

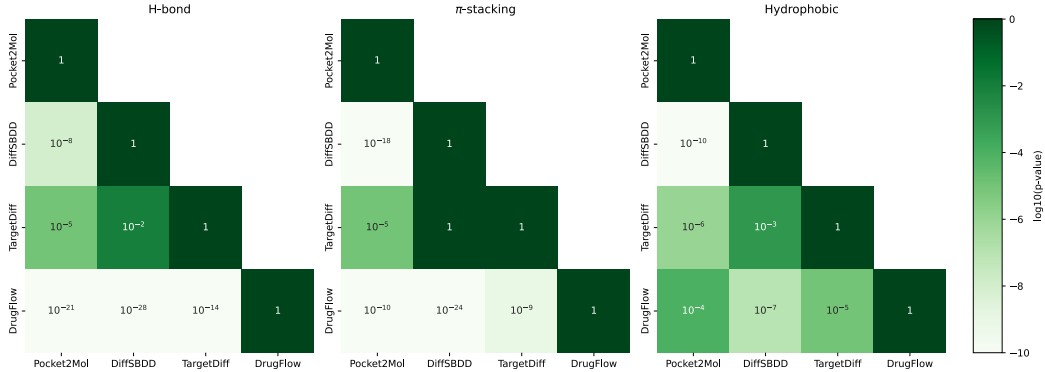

Figure 24: Statistical analysis of Wasserstein distance differences using Student's t-test across normalised numbers of protein-ligand interactions. As in Figure 22, p-values are displayed on a log-scale heatmap, with the last row indicating that distance differences for DRUGFLOW are statistically significant in all cases.

