# OpenReview forum: "Multi-domain Distribution Learning for De Novo Drug Design"
_ICLR.cc/2025/Conference — ICLR 2025 Poster_

### Official Review · Reviewer_67Pj · 2024-10-31

**Soundness:** 3
**Presentation:** 4
**Contribution:** 2
**Rating:** 6
**Confidence:** 3

**Summary:**

This paper describes DrugFlow, a diffusion model for drug-like molecules in 3D. Contributions over existing work are 1) uncertainty estimates from diffusion model 2) an adaptive size selection method 3) protein conformation sampling module 4) a preference alignment optimization scheme.

**Strengths:**

This paper had a lot of strong positives but also some strong negatives. Starting with the positives:

- Good knowledge of the field: unlike many ML papers in this area, this work has no statements about drug discovery that seemed to portray an embarrassing lack of domain knowledge on behalf of the authors. I also agree with the assessment that many works train models for distribution matching and then evaluate them for optimization, which does not make sense
- The end-to-end uncertainty estimate is a really nice idea (even though it isn't clear that it works well, see below)
- The ability to add or delete models during generation is a nice idea and seems to work reasonably well
- I liked that the authors tested a wide range of tasks in the experiments section
- Presentation of the paper is really good, definitely in the top 10%. Regardless of the other criticisms I raise below, I can tell that the authors crafted the manuscript very well

**Weaknesses:**

In my opinion, the biggest weaknesses of this paper all come from the experiments. I've organized them under the following headings

### You might not be measuring the right things

Essentially all metrics in the paper are about how well the distribution of molecules generated by the model matches the training distribution. However:

- Only _marginal_ (1D) distributions seem to be measured, rather than _joint_ distributions of properties (i.e. does the joint distribution of SAscore and logP look the same between training and test). In general, matching the marginal distribution _does not_ imply that the joint distribution matches. Figure 4 is an exception to this.
- QED/logP/SA are all very simple quantities which _do not depend on the 3D structure in any way_. The significance of matching these values is not very clear to me.

### Complete disregard for statistical variation => significance of results is unclear

Almost every quantity estimated is estimated from a finite sample of generated molecules (including Wasserstein distances, JS-distances, coverage of chemical space). This means that all quantities in tables are _statistical estimates_ with finite-sample variation. Moreover, there is additional variation due to the randomness of model training, etc. This variation is not accounted for in any of the Tables (as far as I can tell), making the claims of performance differences poorly supported. I think the paper needs to include measures of variation and/or statistical significance tests to qualify its claims.

### Significance aside, performance over the baselines is unclear

First, the performance of all baselines presumably varies with training. Presumably, more training would give a closer match. Did the authors re-train the baseline models themselves or use a pre-trained checkpoint? Were all models trained a similar amount? Even if the results are statistically significant for a given pair of models, I think it would help to know how much these differences change with training. Perhaps include a plot or table showing how performance changes with training size? (not a specific request, feel free to provide a similar but different piece of evidence if you think it is more appropriate)

Second, it would be helpful to include more baseline models, particularly ones using 2D (ie graph-based) methods. A simple 2D method could be a random perturbation of the SELFIES string for a molecule in the training set. Another option could be an RNN trained on SMILES strings. For any metrics that require 3D coordinates, they could be generated using force fields from rdkit (or some comparable method). I would not expect these baselines to be state-of-the-art, but knowing the performance of simpler methods helps contextualize the performance of more complex ones.

Thirdly, it seems that no baselines were run for the preference optimization experiment?

Finally, it is unclear that the OOD performance described in section 3.2 is _practically_ significant. The uncertainty estimates seem to vary by only a tiny amount (I see that the color bar in Figure 2B has a range of only 0.008). This seems to suggest that the model is not actually very well-calibrated?

**Questions:**

See weaknesses above. The common thread between the weaknesses is answering the question "how well does the model work". Anything the authors can do to answer this question would be helpful.

---

> ### Author Response · Authors · 2024-11-18
> **Response to reviewer 67Pj (1/2)**
>
> We thank the reviewer for their thorough reading, the generally positive assessment of our work and especially for the constructive feedback.
> We agree that significance of results and contextualisation of performance differences are crucial. Following the reviewer’s suggestions, we made an effort to improve the paper in this regard, and provide point-by-point responses below.
> We would be happy to follow up with the reviewer and to further improve our evaluation if necessary.
>
> >**W1: Only marginal (1D) distributions seem to be measured, rather than joint distributions of properties (i.e. does the joint distribution of SAscore and logP look the same between training and test). In general, matching the marginal distribution does not imply that the joint distribution matches. Figure 4 is an exception to this.**
>
> We thank the reviewer for this useful remark and fully agree that matching marginal distributions does not necessarily imply that the joint distribution is identical. To address this concern, we added one more score in Table 1 which compares the joint distributions of four molecular properties ($\sf JSD_{all}$). We would also like to note that Fréchet ChemNet Distance is another method that allows us to compare complex molecular distributions in a general way. Together FCD and $\sf JSD_{all}$ prove that DrugFlow clearly outperforms other methods in the ability to learn the complex and multifaceted molecular distribution. We added the discussion of these results to Section 3.1 (lines 305-309).
>
> >**W2: QED/logP/SA are all very simple quantities which do not depend on the 3D structure in any way. The significance of matching these values is not very clear to me.**
>
> Since assessing the distribution learning capabilities of our model directly in the complex chemical space is intractable, we aim to cover as many orthogonal characteristics of the molecular data as possible. To this end, our evaluation suite encompassess metrics in three major categories: (1) interactions (e.g. Vina score and non-covalent bonds), (2) 3D structure (e.g. bond distances and bond angles), and (3) intrinsic properties of the molecule (e.g. QED, SA, etc.). While admittedly none of these metrics alone suffices to prove the functionality of our model, all of them should be similar to the values of real molecules if the model successfully matches the data distribution. The specific choice of these metrics was inspired by their relevance in downstream drug discovery applications.
>
> >**W3: Almost every quantity estimated is estimated from a finite sample of generated molecules (including Wasserstein distances, JS-distances, coverage of chemical space). This means that all quantities in tables are statistical estimates with finite-sample variation. Moreover, there is additional variation due to the randomness of model training, etc. This variation is not accounted for in any of the Tables (as far as I can tell), making the claims of performance differences poorly supported. I think the paper needs to include measures of variation and/or statistical significance tests to qualify its claims.**
>
> We thank the reviewer for this comment. We performed the Mann-Whitney U test to determine whether the differences between distributions are statistically significant. We compared distributions of scores reported in Tables 1, 2, and Figure 5 for all possible pairs of methods. As shown in the (new) Figures 12-14, and 20, the differences are statistically significant in most of the cases.
>
> >**W4: First, the performance of all baselines presumably varies with training. Presumably, more training would give a closer match. Did the authors re-train the baseline models themselves or use a pre-trained checkpoint? Were all models trained a similar amount? Even if the results are statistically significant for a given pair of models, I think it would help to know how much these differences change with training. Perhaps include a plot or table showing how performance changes with training size? (not a specific request, feel free to provide a similar but different piece of evidence if you think it is more appropriate)**
>
> We followed the reviewers advice and evaluated DrugFlow at different training steps. The results in the (new) Tables 10 and 11 indicate that performance indeed varies with training and is generally improving (Validity, FCD, Rings, RB), as can be expected. Some other properties such as atom types and bond types and geometries (distances and angles) are however learned rather quickly, and more training alone does not seem to guarantee better results. In general, we selected the epoch based on the validity of the samples measured on the validation set. We added this discussion to B.3. For the baselines, we used publicly available models provided by the authors and did not retrain them ourselves under the assumption that the authors provided their best models.

---

> > ### Author Response · Authors · 2024-11-18
> > **Response to reviewer 67Pj (2/2)**
> >
> > >**W5: Second, it would be helpful to include more baseline models, particularly ones using 2D (ie graph-based) methods. A simple 2D method could be a random perturbation of the SELFIES string for a molecule in the training set. Another option could be an RNN trained on SMILES strings. For any metrics that require 3D coordinates, they could be generated using force fields from rdkit (or some comparable method). I would not expect these baselines to be state-of-the-art, but knowing the performance of simpler methods helps contextualize the performance of more complex ones.**
> >
> > We thank the reviewer for this useful suggestion. To address this point we have downloaded about 2.4 million molecules from the ChEMBL database, randomly selected 100 of them for each test set pocket and docked them with Gnina. This is representative of a strong (all molecules are real) unconditional 2D baseline. The results are included in the ablation studies in Tables 7-9. This new baseline helps us consolidate two important findings:
> > 1. DrugFlow matches the training distribution well. The distributions of high-level molecular properties are matched better by DrugFlow than the ChEMBL baseline because ChEMBL molecules are not taken from the same reference set (CrossDocked). This is not true for other generative models. For instance, the FCD metric of other baselines in Table 3 is still worse than that of ChEMBL molecules despite the different reference distributions. DrugFlow on the other hand achieves substantially better results.
> > 2. Pocket-conditioning is important. The ChEMBL baseline underperforms on almost all interaction-related metrics as molecules were assigned without pocket information whereas DrugFlow creates molecules in a pocket-conditioned manner.
> >
> > A discussion of these findings has been added in Appendix B.3 as well.
> >
> > >**W6: Thirdly, it seems that no baselines were run for the preference optimization experiment?**
> >
> > Thank you for this comment. We agree that contextualising the reported performance gains is useful for a better comparison. The primary objective of DPO is to optimise a base model, that was initially trained for distribution learning, toward specific metrics. As such, the reference model serves as the primary baseline for our comparisons.
> >
> > As a simple baseline we report results for a fine-tuned model optimised for the same metrics (Figure 5 and Table 12). Moreover, we have newly added results from two recent related studies (DecompDPO [1] and AliDIFF [2]) in Table 12. For better comparison, we added the Vina score (after local energy minimization) as an additional metric, for which our model was not directly optimised.
> >
> > >**W7: Finally, it is unclear that the OOD performance described in section 3.2 is practically significant. The uncertainty estimates seem to vary by only a tiny amount (I see that the color bar in Figure 2B has a range of only 0.008). This seems to suggest that the model is not actually very well-calibrated?**
> >
> > We thank the reviewer for pointing this out. During training, the model learned to utilise only a small range of uncertainty scores, which we agree limits its interpretability in practice. To address this, we apply a linear transformation to map the scores to the range [0, 1] as shown in (new) Figure 15 and described in Appendix B.4. We updated Figure 2 accordingly.
> >
> > Regarding the question about calibration, Figure 2B aims to demonstrate the sensitivity of the uncertainty score. Even within the data distribution the score is able to detect the main modes and tails, but this is more apparent with an adjusted colour scale for plotting. We added a new figure (Figure 16) using the full colour range for completeness. Although the contrast is worse when using the full range, the differences between modes and tails of the distributions remain visible. The most extreme uncertainty values are only assigned to outliers.
> >
> >
> > **References:**
> >
> > [1] Cheng, X., Zhou, X., Yang, Y., Bao, Y., & Gu, Q. (2024). Decomposed direct preference optimization for structure-based drug design. arXiv. https://arxiv.org/abs/2407.13981
> >
> > [2] Gu, S., Xu, M., Powers, A., Nie, W., Geffner, T., Kreis, K., Leskovec, J., Vahdat, A., & Ermon, S. (2024). Aligning target-aware molecule diffusion models with exact energy optimization. arXiv. https://arxiv.org/abs/2407.01648

---

> > > ### Comment · Reviewer_67Pj · 2024-11-22
> > > **Follow-up questions**
> > >
> > > Thanks for your responses and sorry for the delay responding- it's been a busy week.
> > >
> > > Overall I am happy with the changes made in the paper, but some issues remain unresolved in my opinion. In detail:
> > >
> > > - 1D distributions: measuring multi-dimensional JSD is a good start. However, I am concerned that the methodology is incorrect: you estimate based on a histogram with 10 bins. However, 10 bins for 3 dimensions gives 1000 total bins. With only 100 samples you cannot have a sample in every bin, causing JSD to be infinity unless the distributions filled the *exact* same bins, no? How is it that you attained a finite JSD: do the distributions just happen to overlap almost perfectly?
> > > - QED/logP/SA: I'm fine with your response, but I think joint density matching between these properties *and* vina scores woudl be better (see point above)
> > > - Statistical tests: thanks for adding the U-test! While this is a good start, it only shows that the distributions are significantly different from each other. This is not the main claim of the paper, which is that your method's distribution matches the training distribution more closely. _That_ is the hypothesis which I think most urgently needs a statistical test
> > > - Progress as training progresses: thanks for adding this. Is this also at 100 test samples? Regardless, this makes me less convinced that the method performs better than the baselines. Many of the metrics in Table 10 (e.g. atoms, bonds) fluctuate up and down during training, sometimes doing worse than the baselines. The assumption that the weight checkpoint available online is not clear. For example, in Pocket2Mol they say "We trained Pocket2Mol with batch size 8 and initial learning rate 2 × 10−4 and decayed learning rate by a factor of 0.6 if validation loss did not decrease for 8 validation iterations. We validated the model every 5000 training iterations and the number of total training iterations is 475, 000". It did not seem like they cut off training based on the validation loss.
> > > -  Unconditional baseline: thanks for adding that. What would be the numbers for using Crossdocked itself as the unconditional baseline?
> > > - Preference alignment baselines: thanks for this, I'm happy with it
> > > - OOD performance: please undo the normalization, it is confusing and misleading. I am happy that the uncertainty values are correlated with outputs that might reasonably be considered "uncertain", but it is clearly not fulfilling its role of learning the standard deviation of the outputs. Please just describe this in an honest way as a slight limitation.

---

> ### Author Response · Authors · 2024-11-24
> **Response to the follow-up questions (1/2)**
>
> Thank you for reviewing our changes and providing further clarifications.
>
> We noticed that there might have been a misunderstanding that we evaluate our model on 100 test samples in total. We emphasise that **we sampled 100 molecules for each of 100 target proteins resulting in 10000 samples in total**. We additionally note that this setup is conventional for all SBDD generative methods published before.
>
> We addressed your remaining questions to the best of our understanding and provide the detailed responses below. We hope the discussions and results below fully address your remaining questions and concerns! Please let us know if there are any further opportunities to improve the score.
>
> > **Q1: 1D distributions**
>
> In contrast to KL-divergence $D(p\|q)$ that indeed turns to infinity in case of empty bins in $q$, JS-divergence naturally handles zero bins thanks to the mixture of the distributions it uses. More specifically, JS-divergence between distributions $p$ and $q$ is defined as $\text{JSD}(p\|q)=(D(p\|m)+D(q\|m))/2$, where $m=(p+q)/2$ is the aforementioned mixture. KL-divergence is defined as $D(p\|m)=\sum_{x}p(x)\log\frac{p(x)}{m(x)}$. Therefore, if $m(x)=(p(x)+q(x))/2=0$ then $p(x)=0$ as well, and KL-divergence $D(p\|m)$ turns to 0. That is precisely how it is implemented in [SciPy with function `rel_entr`]( https://docs.scipy.org/doc/scipy/reference/generated/scipy.special.rel_entr.html). This allows us to effectively handle empty bins which is of course the case in our example, as we show in the (new) Figure 20.  Each method has between 2600 and 3100 non-empty bins (out of $10^4=10000$ total bins) with 2500 bins populated by all four methods. We also note that we have 10000 sampled molecules in total (100 per test target).
>
> > **Q2: QED/logP/SA**
>
> We added Vina scores to the $\text{JSD}_{\text{all}}$ and recomputed this metric. As well as before, our method significantly outperforms other baselines (Table 1).
>
> > **Q3: Statistical tests**
>
> We added new results estimating the variation of sample-based Wasserstein distances and JS-divergence using bootstrapping: we computed distances for 20 bootstrapped subsamples (non-overlapping), each comprising 500 sampled molecules (i.e. 5 per test target). We then show in (new) Tables 13-15 that the standard deviation of the distances between subsamples is much smaller than the difference of sample means. We further verify the significance in difference between the resulting Wasserstein distances and JS-divergences by performing Student's t-tests (Figures 21-23).
>
> We hope we understood the suggested measurement methodology correctly. Otherwise we kindly ask the reviewer to suggest the particular evaluation way if any concerns remain uncovered in this context.
>
> > **Q4: Progress as training progresses**
>
> We are struggling to understand the concern of the reviewer. All these aspects (training time, batch size, learning rate, early stopping, etc.) are design choices that can be made in many different ways. Using publicly available pre-trained checkpoints is a valid approach which is widely employed in the machine learning community.
>
> However, we performed a study of the training time dependency **of our own model**, as was originally suggested by the reviewer. Regarding these results, we indeed sampled 100 molecules per test set protein, i.e. 10,000 samples overall (as everywhere else). The most likely explanation for the observed fluctuations is that the affected properties are rather basic and can thus be learned quickly, while more training is required to better match more complex metrics, like validity, FCD and ring system distribution. Small fluctuations around the local optimum can be expected during this process. At the same time, the baselines perform well on the simpler metrics and sometimes outperform our model. To avoid drawing conclusions based on single, potentially imperfect metrics, we include many different metrics and show that DrugFlow has the best all-round performance.
>
> If the reviewer would like to see any specific piece of data we haven't provided yet in this context we are happy to do so.
>
> > **Q5: Unconditional baseline**
>
> We added the results for such a baseline in Tables 7-9. As expected, distances for all molecular properties and geometry metrics are the lowest for the baseline, and distances for protein-ligand interactions are comparable with ChEMBL (due to the same docking procedure and random assignment of protein-ligand pairs).

---

> > ### Author Response · Authors · 2024-11-24
> > **Response to the follow-up questions (2/2)**
> >
> > > **Q6: OOD performance**
> >
> > The uncertainty estimate is not trained to learn the standard deviation of the outputs but that of the flow matching error in each step of the trajectory. We then aggregate these per-step uncertainty estimates along the sampling trajectory to obtain an overall score for each atom as described in Section 2.1. **We do not claim the score estimates the standard deviation of the outputs**. Ultimately, we are interested in quantifying the **generative capabilities** of our model, i.e. in the discriminative power of the score to detect OOD – which we demonstrate it does regardless of the range of values it takes. Note that the motivation in appendix A.3 is only a loose justification for the aggregation method. We clarified this important distinction in the relevant Appendix Section A.3. We also undid the normalisation following the reviewer’s advice and mentioned the suggested slight limitation in Section 3.2. We might have misunderstood the reviewer's comment, so we kindly request a more precise description of the mentioned limitation if the concern is still not satisfactorily addressed.
> >
> > ___
> >
> > Regardless of the reviewer’s final decision, we would like to thank the reviewer for their time, interactions and all the thorough comments. The reviewer’s contributions made the evaluation part of this work much stronger and more convincing. Thank you!

---

> > > ### Comment · Reviewer_67Pj · 2024-11-24
> > > **Thank you for answering my followup questions**
> > >
> > > Thank you for clarifying that it is 100 samples for 100 targets. That gives statistical tests a reasonable amount of power. Sorry that I misunderstood this earlier
> > >
> > > Q1: JS divergence. You are completely write about the definition, sorry. I misremembered it as $[KL(q|p) + KL(p|q)] / 2$. When I wrote that comment I was on a plane and did not have access to internet and could not double check that. It's not an excuse though, I'm very sorry. Looks like I have become the kind of reviewer I complain about myself!!! No more issues here.
> > >
> > > Q2: thanks for adding the vina score, I think this helps.
> > >
> > > Q3: thanks for the bootstrap estimate. No further concerns here.
> > >
> > > Q4: progress as training progresses. Thank you for adding what you have added so far. My concern about training time is that all methods will have their distribution matching statistics change over time, and since carefully matching the training distribution is not usually an objective that other papers explicitly optimize for it is possible that the comparison models are under (or over) trained, and selecting a different checkpoint (or using a different number of parameters) would give substantially different results, especially since the absolute differences are (relatively) small.
> > >
> > > Q5: thanks for adding this. I think this is a helpful reference point for the reader.
> > >
> > > Q6: good point, even though I knew the standard deviation is the standard deviation in the score, I was somewhat conflating this with the standard deviation of the output.
> > >
> > > Overall I am happy with the changes. I have a lingering feeling that the method may not really be an advance of capabilities, but I think you have done diligent experiments and have addressed my concerns about correctness. For the time being I will maintain my score as a weak accept, and I will convey this sentiment to the other reviewers. I know that me keeping my score might not seem like a win, but I was considering lowering it earlier and have now decided not too. Thanks for all your hard work!

---

> > > > ### Author Response · Authors · 2024-11-25
> > > > **Thanks for your feedback!**
> > > >
> > > > Thank you for your responsiveness and the generally positive feedback. We really appreciate your collaboration.
> > > > We would like to provide one last clarification regarding Q4 as this seems to be causing your remaining doubts.
> > > >
> > > > In particular, we respectfully disagree with the statement that “matching the training distribution is not usually an objective that other papers explicitly optimize for”. Almost all structure-based molecule generative models (and importantly, all our baselines Pocket2Mol, TargetDiff, and DiffSBDD) are directly trained as generative models on the same training data set. Matching the data distribution is therefore their primary training objective. Nevertheless, most previous works have not extensively evaluated these capabilities in the past and instead focused on absolute values (e.g. best Vina scores) which does not reflect their training objective. That is the main reason behind the proposed evaluation methodology.
> > > >
> > > > > **differences between checkpoints/parameters**
> > > >
> > > > Your comment about the fluctuations of some metrics between different checkpoints is undeniable. As outlined in our previous response, we believe that a certain amount of variability is completely normal in a complex multi-objective evaluation. More importantly, however, we would like to take this opportunity to emphasise that **the vast majority of metrics show consistent improvements over baselines despite the fluctuations**.
> > > >
> > > > Below, we combined the ablation results from Tables 10-12 with the baseline values from Tables 1-3 to more clearly highlight this finding.
> > > >
> > > >
> > > > > **advance of capabilities**
> > > >
> > > > In our personal, admittedly biased, opinion our paper represents an advancement in the correct evaluation of distribution learners for SBDD, and additionally introduces a new model that consistently achieves top results in this framework (sometimes even by a large margin).
> > > > Moreover, we could show how preference alignment can be combined with the strong distribution learner to achieve better absolute values, which are in accordance with the modified training objective.

---

> > > > > ### Author Response · Authors · 2024-11-25
> > > > > **Supplementary Tables**
> > > > >
> > > > > | Baselines/Epochs | FCD | Atoms | Bonds | Rings |
> > > > > | --- | --- | --- | --- | --- |
> > > > > | `Pocket2Mol` | $\text{12.703}$ | $\text{0.081}$ | $\textbf{0.044}$ | $\text{0.446}$ |
> > > > > | `DiffSBDD` | $\text{11.637}$ | $\text{0.050}$ | $\text{0.227}$ | $\text{0.588}$ |
> > > > > | `TargetDiff` | $\text{13.766}$ | $\text{0.076}$ | $\text{0.240}$ | $\text{0.632}$ |
> > > > > ||
> > > > > | `DrugFlow, epoch=100` | $\text{6.064}$ | $\underline{\text{0.044}}$ | $\text{0.070}$ | $\text{0.476}$ |
> > > > > | `DrugFlow, epoch=300` | $\text{5.775}$ | $\text{0.078}$ | $\underline{\text{0.045}}$ | $\underline{\text{0.416}}$ |
> > > > > | `DrugFlow, epoch=500` | $\underline{\text{5.154}}$ | $\text{0.067}$ | $\text{0.063}$ | $\text{0.416}$ |
> > > > > | `DrugFlow, epoch=600` | $\textbf{4.278}$ | $\textbf{0.043}$ | $\text{0.060}$ | $\textbf{0.391}$ |
> > > > >
> > > > >
> > > > > | Baselines/Epochs | C--C | C--N | C=C | C--C=C | C--C--C | C--C--O | QED | SA | logP | Rotatable bonds |
> > > > > | --- | --- | --- | --- | --- | --- | --- | --- | --- | --- | --- |
> > > > > | `Pocket2Mol` | $\text{0.050}$ | $\text{0.024}$ | $\text{0.045}$ | $\text{2.173}$ | $\text{2.936}$ | $\text{3.938}$ | $\text{0.072}$ | $\text{0.576}$ | $\text{1.209}$ | $\text{2.861}$ |
> > > > > | `DiffSBDD` | $\text{0.041}$ | $\text{0.039}$ | $\text{0.042}$ | $\text{3.632}$ | $\text{8.166}$ | $\text{7.756}$ | $\text{0.065}$ | $\text{1.570}$ | $\text{0.774}$ | $\text{0.928}$ |
> > > > > | `TargetDiff` | $\text{0.017}$ | $\underline{\text{0.019}}$ | $\text{0.028}$ | $\text{4.281}$ | $\text{3.422}$ | $\text{4.125}$ | $\text{0.050}$ | $\text{1.518}$ | $\text{0.489}$ | $\text{0.354}$ |
> > > > > ||
> > > > > | `DrugFlow, epoch=100` | $\text{0.040}$ | $\text{0.035}$ | $\text{0.025}$ | $\text{1.021}$ | $\text{2.035}$ | $\text{1.811}$ | $\text{0.022}$ | $\text{0.655}$ | $\textbf{0.344}$ | $\text{0.719}$ |
> > > > > | `DrugFlow, epoch=300` | $\textbf{0.015}$ | $\text{0.020}$ | $\textbf{0.014}$ | $\textbf{0.508}$ | $\underline{\text{1.533}}$ | $\textbf{1.372}$ | $\text{0.055}$ | $\text{0.589}$ | $\text{1.260}$ | $\text{0.515}$ |
> > > > > | `DrugFlow, epoch=500` | $\underline{\text{0.016}}$ | $\text{0.024}$ | $\text{0.018}$ | $\underline{\text{0.883}}$ | $\textbf{1.417}$ | $\underline{\text{1.417}}$ | $\underline{\text{0.014}}$ | $\underline{\text{0.438}}$ | $\underline{\text{0.487}}$ | $\underline{\text{0.347}}$ |
> > > > > | `DrugFlow, epoch=600` | $\text{0.017}$ | $\textbf{0.016}$ | $\underline{\text{0.016}}$ | $\text{0.952}$ | $\text{2.269}$ | $\text{1.941}$ | $\textbf{0.014}$ | $\textbf{0.317}$ | $\text{0.665}$ | $\textbf{0.144}$ |
> > > > >
> > > > > | Baselines/Epochs | Vina | Gnina | H-bond | H-bond (acc.) | H-bond (don.) | $\pi$-stacking | Hydrophobic |
> > > > > | --- | --- | --- | --- | --- | --- | --- | --- |
> > > > > | `Pocket2Mol` | $\text{0.064}$ | $\text{0.044}$ | $\text{0.040}$ | $\text{0.026}$ | $\text{0.014}$ | $\text{0.007}$ | $\textbf{0.027}$ |
> > > > > | `DiffSBDD` | $\text{0.086}$ | $\text{0.043}$ | $\text{0.047}$ | $\text{0.030}$ | $\text{0.017}$ | $\text{0.011}$ | $\text{0.044}$ |
> > > > > | `TargetDiff` | $\underline{\text{0.034}}$ | $\text{0.030}$ | $\text{0.031}$ | $\text{0.021}$ | $\text{0.010}$ | $\text{0.012}$ | $\text{0.039}$ |
> > > > > ||
> > > > > | `DrugFlow, epoch=100` | $\text{0.038}$ | $\underline{\text{0.019}}$ | $\text{0.028}$ | $\text{0.016}$ | $\text{0.012}$ | $\text{0.007}$ | $\underline{\text{0.028}}$ |
> > > > > | `DrugFlow, epoch=300` | $\text{0.039}$ | $\text{0.035}$ | $\textbf{0.019}$ | $\textbf{0.011}$ | $\underline{\text{0.008}}$ | $\textbf{0.004}$ | $\text{0.075}$ |
> > > > > | `DrugFlow, epoch=500` | $\text{0.056}$ | $\text{0.029}$ | $\text{0.024}$ | $\underline{\text{0.011}}$ | $\text{0.013}$ | $\underline{\text{0.005}}$ | $\text{0.041}$ |
> > > > > | `DrugFlow, epoch=600` | $\textbf{0.028}$ | $\textbf{0.013}$ | $\underline{\text{0.019}}$ | $\text{0.012}$ | $\textbf{0.007}$ | $\text{0.006}$ | $\text{0.036}$ |

---

> > > > > ### Comment · Reviewer_67Pj · 2024-11-25
> > > > > **Final comment**
> > > > >
> > > > > I think the review period will allow me to make just one final comment, so let me respond to your points above.
> > > > >
> > > > > **Distribution matching as an objective**: you are completely right that matching the training distribution _is_ an objective that other papers explicitly optimize for. I wrote that comment quickly and it was poorly worded. What I meant is that other methods are usually not extensively evaluating their degree of distribution matching and don't choose optimize their training setup to get a really good match (eg stopping training at a carefully chosen time). In your experiments you do indeed show that despite the fluctuations, the values tend to be lower than the baselines. However, I think it is still possible that the baselines would achieve better matching if training continued. This is something I suggest you test for the camera-ready.
> > > > >
> > > > > **Advancement**: you make a good point- I do think that this paper makes an advancement in _evaluation_. Thanks for this!

---

> ### Author Response · Authors · 2024-11-26
> **Thank you!**
>
> We thank the reviewer for the clarification. We agree that exploring how other baselines can be further improved is an interesting question that however exceeds the scope of a single conference publication. We will take this recommendation into account and thank the reviewer again for the comprehensive feedback and their contribution to our work!

---

### Official Review · Reviewer_o8Zm · 2024-11-02

**Soundness:** 3
**Presentation:** 3
**Contribution:** 3
**Rating:** 8
**Confidence:** 4

**Summary:**

This paper proposes DrugFlow, a multi-modal flow matching model for structure-based drug design. DrugFlow jointly models the distribution of ligand structures and receptor sidechain structures. It also includes an uncertainty estimate module and an adaptive ligand size selection module that address the issues overlooked by previous work. In addition, preference alignment techinique is used for property optimization, which increases the value of this work. Overall, this is a nice work that orchestrated various machine learning techniques, which are all well justified in the context.

**Strengths:**

- The model considers side-chain flexibility, which is critical in ligand docking and design as receptors are mostly non-rigid. The side-chain flexibility issue has also been overlooked in previous SBDD methods until this work, to the best of my knowledge.
- This model provides an estimate of uncertainty, which is improtant in molecular modeling area and can increase the practicality of the method. Uncertainty estimation has been a common practice in structure prediction settings, but it has also been overlooked in the previous structure-based drug generation methods.
- This work demonstrated the use of preference alignment to control the properties of generated molecules, which increases the value of the model as in SBDD, there are many properties other than receptor structures need to be considered.
- All the techniques introduced in this work are well justified by clearly organized experiments (Section 3.2-3.5). Notably, uncertainties visualized in Figure 2 are very informative and I find unrealistic structures (long carbon chains with bifurcation) were assigned high uncertainties, which agrees with the intuition that such unrealistic structures are uncommon in the dataset.

**Weaknesses:**

- Does the evaluation presented in Section 3.1 consider side-chain flexibility? It seems that the DrugFlow and FlexFlow are separate variants and only the FlexFlow considers side-chain flexiblity.
- If Section 3.1 does not model side-chain flexibility, why not? Did the authors consider jointly sampling both ligand structures and side-chain torsional angles?

**Questions:**

See Weaknesses

---

> ### Author Response · Authors · 2024-11-18
> **Response to reviewer o8Zm**
>
> Thank you for the positive feedback. We address your two remaining questions below.
>
> >**W1: Does the evaluation presented in Section 3.1 consider side-chain flexibility? It seems that the DrugFlow and FlexFlow are separate variants and only the FlexFlow considers side-chain flexiblity.**
>
> This observation is correct. Our base model, DrugFlow, keeps the entire pocket structure (including side chains) fixed and Section 3.1 indeed only discusses this variant.
>
> >**W2: If Section 3.1 does not model side-chain flexibility, why not? Did the authors consider jointly sampling both ligand structures and side-chain torsional angles?**
>
> FlexFlow is an extension of our base model that samples side chain torsion angles _in addition to the ligand structure_ (atom types, bond types and coordinates). Since the benchmark in Section 3.1 includes baselines that all treat the side chain torsion angles as fixed, we decided to focus this comparison on the DrugFlow model and discuss FlexFlow separately in Section 3.4. However, we also report all key metrics for FlexFlow as part of our ablation study (Appendix B.3, Tables 7-9). Generally, FlexFlow achieves comparable results to DrugFlow despite the additional complexity of the task.

---

### Official Review · Reviewer_6xpR · 2024-11-04

**Soundness:** 1
**Presentation:** 3
**Contribution:** 2
**Rating:** 6
**Confidence:** 5

**Summary:**

The paper introduces DRUGFLOW, a generative model designed for structure-based drug design. It seamlessly combines continuous flow matching with discrete Markov bridges to capture the chemical, geometric, and physical characteristics of three-dimensional protein-ligand data. The model provides an uncertainty estimate to detect out-of-distribution samples and employs a joint preference alignment strategy to guide sampling towards desirable metric values. Furthermore, the paper extends the model to explore protein conformational landscapes by concurrently sampling side chain angles.

**Strengths:**

- The paper is articulated clearly and concisely.
- Figures and tables effectively present complex data and comparisons, enhancing accessibility for readers.
- The methodology is detailed thoroughly, supporting reproducibility.

**Weaknesses:**

The primary concern lies in the paper's technical soundness:

1. The treatment of the pocket is inadequately detailed—it is unclear whether the pocket is generated jointly with the molecule or used as context.
2. In Section 2.1, the uncertainty estimation involves several ambiguities:
   - In line #133, the assumption of the error being normally distributed is neither evident nor justified.
   - In line #143, $\dot{x}_t$ is inaccurately referred to as a ground truth vector field; it should be considered a conditional vector field, given $x_0$ is known.
   - In line #987, $\underset{\theta}{\max} $ is mistakenly used instead of argmax; also, maximizing Equation 30 is not equivalent to minimizing Equation 31, despite sharing the same minima—the loss surfaces differ.
   - The described technique resembles regularization, yet its role in quantifying an atom's uncertainty score remains unclear.
3. In Section 2.2, the concept of a virtual node demands more explanation; specifically, it's unclear whether virtual bonds exist when virtual nodes are incorporated.

**Questions:**

1. FlexFlow samples side chain configurations additionally—does pocket data exist pre- and post-binding? If absent, how does this approach differ from treating the pocket as context?
2. In line #134, on what basis is the error assumed to be normally distributed?
3. During training, while virtual nodes are added to each sample, are virtual bonds similarly included?
4. How does the paper synthesize preference pairs, as noted in line #206?
5. Why does the author evaluate bonds using the Wasserstein distance in Table 1, whereas other studies [1] and [2] apply KL and Jensen-Shannon divergences?
6. Why does the reported Wasserstein distance for QED, SA, and logP differ from those in Table 1 of paper [3]?

[1] Xingang Peng, Shitong Luo, Jiaqi Guan, Qi Xie, Jian Peng, Jianzhu Ma, "Pocket2Mol: Efficient Molecular Sampling Based on 3D Protein Pockets"
[2] Jiaqi Guan, Wesley Wei Qian, Xingang Peng, Yufeng Su, Jian Peng, Jianzhu Ma. "3D Equivariant Diffusion for Target-aware Molecule Generation and Affinity Prediction"
[3] Arne Schneuing1, Charles Harris, Yuanqi Du, Kieran Didi, Arian Jamasb, Ilia Igashov, Weitao Du, Carla Gomes, Tom L. Blundell, Pietro Lio, Max Welling, Michael Bronstein. "Structure-based Drug Design with Equivariant Diffusion Models"

---

> ### Author Response · Authors · 2024-11-13
> **Response to reviewer 6xpR (1/2)**
>
> We thank the reviewer for the critical assessment of our manuscript. We believe to have addressed all the concerns and updated the manuscript accordingly, with detailed responses provided below.  We hope the reviewer increases their score and will be glad to address any further questions or comments.
>
> >**W1: The treatment of the pocket is inadequately detailed—it is unclear whether the pocket is generated jointly with the molecule or used as context.**
>
> We thank the reviewer for this comment. Indeed, DrugFlow is the model that always uses the pocket as a fixed context, and FlexFlow generates pocket side chain angles jointly with the molecule while still keeping amino acid identities and backbone coordinates fixed. Further details on the pocket representation are described in Section 2.3 and Appendix A.5. To clarify this further, we added the sentence “Both DrugFlow and FlexFlow are conditioned on fixed protein backbone coordinates and amino acid types, which are used as context for denoising.” in the introduction (line 84-86).
>
> >**W2: In Section 2.1, the uncertainty estimation involves several ambiguities:**
> >- **In line #133, the assumption of the error being normally distributed is neither evident nor justified.**
>
> The Gaussian error distribution is indeed a modelling assumption (and thus not derived). It is very commonly used in related literature [4,5,6]. Even though this choice was not initially motivated by empirical observations, we added an illustrative plot of the regression error derived from one training batch of DrugFlow to demonstrate that the empirical error distribution closely resembles a Gaussian one (Figure 21). We believe this empirical evidence better justifies our modelling assumption and we thank the reviewer for their remark.
>
> >- **In line #143, $\dot{x}_t$ is inaccurately referred to as a ground truth vector field; it should be considered a conditional vector field, given $x_0$ is known.**
>
> We agree with this comment and added “conditional” to this line to reduce the ambiguity of the formulation.
>
> >- **In line #987, $\max_\theta$ is mistakenly used instead of argmax; also, maximizing Equation 30 is not equivalent to minimizing Equation 31, despite sharing the same minima—the loss surfaces differ.**
>
> We agree with this comment and replace the max operations with argmax, which allows us to omit the constant without any problems. We additionally expand the transition from (30) to (32) in more detail to demonstrate its correctness and remove the term “equivalent” to avoid any ambiguities.
>
> >- **The described technique resembles regularization, yet its role in quantifying an atom's uncertainty score remains unclear.**
>
> This method has been successfully used as an uncertainty estimate in the past [4,5], and here we empirically demonstrate its utility for out-of-distribution detection in structure-based drug design (Figure 2). The motivation is discussed in Sections 2.1, A.2, and A.3. To summarize the intuition behind the technique, the model estimates the variance of its own predictions in addition to the mean (the most likely value). If this variance is high, the model “believes” the true value could be far away from the predicted value, i.e. the model is uncertain.
>
> While we would be happy to further improve the clarity of this section, we struggle to understand the concern raised by the reviewer. We kindly ask the reviewer to specify what is unclear and will be happy to address any further questions.
>
> >**W3: In Section 2.2, the concept of a virtual node demands more explanation; specifically, it's unclear whether virtual bonds exist when virtual nodes are incorporated.**
>
> All virtual nodes are disconnected from all other nodes in the graph, i.e. all edges of virtual nodes are assigned the “None” type. We clarified this in Section 2.2 (line 170-171). We will be happy to provide additional clarifications if needed.

---

> ### Author Response · Authors · 2024-11-13
> **Response to reviewer 6xpR (2/2)**
>
> >**Q1: FlexFlow samples side chain configurations additionally—does pocket data exist pre- and post-binding? If absent, how does this approach differ from treating the pocket as context?**
>
> Pocket backbone atoms and amino acid types are provided as input and are not changed during the generation process. Only the side-chain $\chi$-angles are variable and denoised by FlexFlow. We clarified this with a new sentence (lines 84-86). The training data consists of bound pocket poses.
>
> >**Q2-Q3**
>
> see W2 and W3
>
> >**Q4: How does the paper synthesize preference pairs, as noted in line #206?**
>
> Section 2.4 (including the mentioned line) introduces our general method for preference optimization using any preference pairs, while Section 3.5 (lines 450-452) provides a detailed description of what metrics were used and how pairs were assembled. We added a sentence in Section 2.4 that refers to Section 3.5 for clarity.
>
> >**Q5: Why does the author evaluate bonds using the Wasserstein distance in Table 1, whereas other studies [1] and [2] apply KL and Jensen-Shannon divergences?**
>
> We decided to use Wasserstein distance for bond lengths and bond angles because, unlike KL- and JS-divergence, it takes into account the underlying metric space. For a concrete example why this matters, please see https://stats.stackexchange.com/a/351153. If we assume the x-axis of these plots represents bond lengths or bond angles, the distributions on the left are arguably less similar than the distributions on the right, which is correctly captured by the Wassertstein distance but not by the KL-divergence.
>
> We appreciate your comment and are aware that prior works made a different experimental design choice. To account for this, we added a new table providing Jensen-Shannon divergence results for the same quantities (Table 5). Exactly as well as in case of Wasserstein distance, DrugFlow outperforms other methods in all metrics except logP where it performs the second – on par with TargetDiff.
>
> >**Q6: Why does the reported Wasserstein distance for QED, SA, and logP differ from those in Table 1 of paper [3]?**
>
> We used the publicly available code bases to sample molecules and computed the Wasserstein distances using our own evaluation scripts. Small differences could be explained by implementation details. More importantly, however, we use the training set as our reference distribution whereas reference [3] used the substantially smaller test set, which explains the different numerical results.
>
> **References**
>
> [1] Xingang Peng, Shitong Luo, Jiaqi Guan, Qi Xie, Jian Peng, Jianzhu Ma, "Pocket2Mol: Efficient Molecular Sampling Based on 3D Protein Pockets"
>
> [2] Jiaqi Guan, Wesley Wei Qian, Xingang Peng, Yufeng Su, Jian Peng, Jianzhu Ma. "3D Equivariant Diffusion for Target-aware Molecule Generation and Affinity Prediction"
>
> [3] Arne Schneuing1, Charles Harris, Yuanqi Du, Kieran Didi, Arian Jamasb, Ilia Igashov, Weitao Du, Carla Gomes, Tom L. Blundell, Pietro Lio, Max Welling, Michael Bronstein. "Structure-based Drug Design with Equivariant Diffusion Models"
>
> [4] Lakshminarayanan, Balaji, Alexander Pritzel, and Charles Blundell. "Simple and scalable predictive uncertainty estimation using deep ensembles." Advances in neural information processing systems 30 (2017).
>
> [5] Nix, David A., and Andreas S. Weigend. "Estimating the mean and variance of the target probability distribution." Proceedings of 1994 ieee international conference on neural networks (ICNN'94). Vol. 1. IEEE, 1994.
>
> [6] Kalman, Rudolph Emil. "A new approach to linear filtering and prediction problems." (1960): 35-45.

---

### Author Response · Authors · 2024-11-18
**Response to all reviewers**

We thank all the reviewers for considering our manuscript and providing valuable feedback.
In the updated version of the paper, we addressed all concerns raised. Most importantly,
- We corrected all ambiguities identified by reviewer 6xpR,
- We report a new metric ($\text{JSD}_\text{all}$) that assesses the match of the joint distribution of several properties,
- We conducted pairwise statistical tests and demonstrated that differences between methods are significant, and
- We provide new baselines that help contextualise our results.

Changes are highlighted with a teal colour.
Furthermore, we answered questions and provided further clarifications in the point-by-point responses below.

---

### Meta-Review · Area_Chair_ZdxZ · 2024-12-20

**Metareview:**

This paper integrates continuous flow matching and discrete Markov bridges to form a generative model for structure-based drug design. It empirically demonstrates performance gains of the aligned and fine-tuned models compared to the training data and the reference model. The reviewers have found this work interesting and suitable for publication at ICLR.

**Additional Comments On Reviewer Discussion:**

During the rebuttal phase, the authors have addressed the concerns that reviewer 6xpR had about the technical soundness of the paper via a thorough discussion and revisions to the paper.

---

### Decision · Program_Chairs · 2025-01-22

Accept (Poster)